

# A physically-based correction for stray light in Brewer spectrophotometer data analysis

Vladimir Savastiouk[1,‡], Henri Diémoz[2,‡], and C. Thomas McElroy[3,⋆]

[1]International Ozone Services Inc., Toronto, Ontario, Canada
[2]Regional Environmental Protection Agency (ARPA) of the Aosta Valley, Saint-Christophe, Italy
[‡]These authors contributed equally to this work.
[3]Department of Earth and Space Science and Engineering, York University, Toronto, Ontario, Canada
[⋆]retired

**Correspondence:** V. Savastiouk (vl@dimir.ca)

**Abstract.** Brewer ozone spectrophotometers have become an integral part of the global ground-based ozone monitoring network collecting data since the early 1980s. The double-monochromator Brewer version (MkIII) was introduced in 1992. With the Brewer hardware being so robust, both single- and double-monochromator instruments are still in use. The main difference between the single Brewers and the double Brewers is the much lower stray light in the double instrument. Laser scans estimate the rejection level of the single Brewers to be $10^{-4.5}$ while the doubles improve this to $10^{-8}$, virtually eliminating the effects of stray light. For a typical single-monochromator Brewer, stray light leads to an underestimation of ozone of approximately 1 % at 1000 DU ozone SCD and can exceed 5 % at 2000 DU, while underestimation of sulphur dioxide reaches 30 DU when no sulphur dioxide is present. This is because even a small additional stray light at shorter wavelengths significantly reduces the slant ozone at large values. An algorithm for stray light correction based on the physics of the instrument response to stray light which adds light from longer wavelengths to shorter ones has been developed. The simple assumption is that count rates measured at any wavelength have a contribution from stray light from longer, and thus brighter, wavelengths because of the ozone cross-section gradient leading to a rapid change in intensity as a function of wavelength. Using the longest measured wavelength (320 nm) as a proxy for the overall brightness provides an estimate of this contribution. The sole parameter, in the order of 0.2 to 0.6 %, that describes the percentage of light at the longest wavelength to be subtracted from all channels is determined by matching ozone calculations from the single and the double-monochromator Brewers. Removing this additional count rate from the signal mathematically before deriving ozone corrects for the extra photons scattering within the instrument that produces the stray light effect. Analysing historical data from co-located single and double-monochromator Brewers can provide an estimate of how the stray light contribution changes over time in an instrument. The corrected count rates of the measured wavelengths can also be used to improve other calculations: the sulphur dioxide column, the aerosol optical depth, the effective temperature of the ozone layer or any other products. Also presented, is an initial analysis of signs consistent with the stray light effect in the double-monochromator Brewers. A multi-platform code to correct the count rates for stray light and saving the corrected values in a new B-file for use with any existing Brewer data analysis software is available to the global Brewer user community at https://zenodo.org/record/8097039 (Savastiouk and Diémoz, 2023).



## 1 Introduction

Developed in the late 1970s at what is now known as Environment and Climate Change Canada (ECCC), the Brewer spectrophotometer has become an important research and monitoring ground-based instrument to determine, among other quantities, the amount of ozone in the atmosphere and the solar ultraviolet (UV) irradiance reaching the surface (Kerr et al., 1985). As part of the Network for the Detection of Atmospheric Composition Change (NDACC), the Brewer network, which today consists of more than 200 operating instruments, contributes with its long-term data set to understanding the chemistry and dynamics of the Earth's atmosphere, and their variations due to anthropogenic emissions and climate change (Eleftheratos et al., 2022; WMO, 2022; Petkov et al., 2023).

The first Brewers were conceived as single-monochromator spectrometers, i.e. they were equipped with only one dispersion element to separate sunlight into a range of individual wavelengths to be analysed. Double-monochromator Brewers were introduced only in the 1990s to achieve better spectral purity, thus reducing the so-called internal spectral (or out-of-band) stray light. The latter is defined as the fraction of radiation with wavelengths other than the one being measured, and yet able to reach the detector due to scatter inside the instrument. Spectral stray light is not peculiar to the Brewer spectrophotometer, but to any single-monochromator system, including new-generation instruments based on array detectors (Zong et al., 2006). Indeed, the problem is actually worse for array detectors because of the much larger solid angle within the instrument that contributes to the signals measured. There are other types of stray light, e.g. originated from scattering of sky light entering the instrument field of view or from ineffective filtering of polarised light, but they are not addressed in the present study.

Due to the strong irradiance gradient in the UV-B band of the solar spectrum, spectral stray light makes irradiance at shorter wavelengths seem significantly higher than it actually is, because of photons with longer wavelengths being detected when observing short wavelengths. This instrumental artefact leads to an overestimation of the global irradiance in the UV-B (wavelength range 280–315 nm, Lantz et al., 2002) and underestimation of total ozone and sulphur dioxide (e.g., Redondas et al., 2014). Other retrieved quantities, such as the ozone profile (Petropavlovskikh et al., 2011) or the aerosol optical depth (AOD, Arola and Koskela, 2004; Silva and Kirchhoff, 2004; Hrabčák, 2018; López-Solano et al., 2018) may be affected as well. The contribution to the measured radiation from stray light is generally small in an absolute sense, however as the amount of ozone, and thus the absorption, increases, the true sunlight gets very dim, while stray light stays relatively bright. This is especially true for measurements at large solar zenith angles at times close to sunrise and sunset, or at high latitudes, where the slant column density (SCD, i.e. air mass times the vertical column) of ozone is large most of the year due to both solar elevation and ozone climatology. However, stray light may complicate ozone calculations even in mid-latitudes in the winter (Stübi et al., 2017).

With the Brewer hardware being so robust, both the single and the double-monochromator instruments are still in use. Therefore, to make retrievals from single Brewers comparable to double Brewers, the effect of spectral stray light must be thoroughly



characterised and corrected. Only then can the global ozone data set be homogenised to allow an accurate determination of long-term trends or the validation of estimates made from the space.

Several studies have addressed the effect of stray light on Brewer retrievals and proposed some solutions going beyond the mere cutting off of measurements at large solar zenith angles (Stübi et al., 2017). These algorithms often rely on complex parameterisations to correct the calculated ozone vertical column density (VCD) as a function of the optical path through the atmosphere (Karppinen et al., 2015; Redondas et al., 2018; Vaziri Zanjani et al., 2019). Only a few authors proposed methods making direct use of the light detected by the Brewer. Kerr (2002), for example, derived the stray light spectrum based on actual measurements, however the methodology relies on a non-standard (group-scan) routine and on a specific (laser-based) instrumental characterisation not usually available. Hrabčák (2018) derived an approximate estimate of the stray light effect from global UV measurements on cloudless days, then parameterised it as a polynomial function of the solar zenith angle to provide more accurate AOD values.

This study presents a much simpler algorithm (PHYSically-based Correction for Stray light, PHYCS) to make the Brewer retrievals of total ozone and sulphur dioxide virtually unaffected by stray light. The effect on global UV irradiance measurements is not addressed here. Our algorithm is based on reversing the stray light effect mathematically by subtracting its contribution from the detected count rates before calculations for any products are done. PHYCS only needs calibration of one parameter for ozone and one for sulphur dioxide, without requiring peculiar instrumental characterisations or complex parameterisations. This method has several advantages:

- It only uses the count rates detected by the Brewer to estimate and remove the stray light contribution. Hence, the effects of the instrument characteristics are fully separated from the effects of the environmental conditions, so that the algorithm can be applied as it is to any observation geometry;

- We implemented it in a multi-platform software package. The latter corrects the count rates in the original data files, thus it allows the operators to keep using any existing Brewer data processing tools for the constituent retrievals;

- It can readily become part of the calibration process during regular on-site audits with travelling standards, provided that the latter ones are already characterised for the stray light effect.

The present paper is organised as follows. Section 2 introduces PHYCS and its general principles. Section 3 presents the results of radiative transfer calculations, confirming the assumptions which are at the basis of the algorithm, and an application to the analysis of real data. A discussion about the implications for the Brewer calibration and data processing follows (Sect. 4). Conclusions and perspectives complete the article (Sect. 5).





**Table 1.** Brewer spectrophotometers employed in this study.

| Brewer no. | Type | Agency | Location | Coordinates and altitude |
|---|---|---|---|---|
| #009 | MkIV | NOAA | Mauna Loa, Hawaii (USA) | 19.5° N, 155.5° W, 3397 m a.s.l. |
| #119 | MkIII | | | |
| #008 | MkII | ECCC | Toronto, Ontario (Canada) | 43.8° N, 79.5° W, 150 m a.s.l. |
| #145 | MkIII | | | |
| #029 | MkII[a] | ECCC | Alert, Nunavut (Canada) | 82.5° N, 62.5° W, 20 m a.s.l. |
| #246 | MkIII | | | |
| #017 | MkII | ECCC | Travelling standard | Travelling standard |
| #109 | MkIV | IOS | Travelling standard | Travelling standard |

[a] Brewer #029 is actually a MkV, i.e. a Brewer with a modified filter wheel to be able to switch from UV to visible. However, this is not relevant for the present study.

## 2 Data and methods

### 2.1 Instruments and Data

The Brewer spectrophotometer and the standard data processing algorithm are described in numerous publications (see, for example, Siani et al. (2018), and Zhao et al. (2021) and references therein). In Appendix A, we provide a basic background only as needed for this paper.

To show that PHYCS works well under different atmospheric conditions, data from the following Brewers at 3 locations were analysed:

- Brewers #009 and #119 at Mauna Loa Observatory (MLO) in Hawaii, USA. This station is a pristine, premium site for Langley calibrations, located in the tropics at high elevation. Low total ozone and little ozone variability in the seasons are expected at this location. The instruments, belonging to ECCC, are operated jointly by the National Oceanic and Atmospheric Administration (NOAA) and ECCC;

- Brewers #008 and #145 in Toronto, Ontario, Canada. The station is in an urban environment located at midlatitude at low elevation. Moreover, the Toronto station is also home for the world Brewer reference (Zhao et al., 2021). The instruments are operated by ECCC;

- Brewers #029 and #246 at Alert, Nunavut, Canada. The station is located on the coast of an island surrounded by ice all-year-round, in a high latitude environment. Alert is thus characterised by high variability in ozone VCD throughout the year. The range of the available solar elevation angles is limited owing to the high latitude of the site. The Brewers are operated by ECCC.





**Table 2.** Stations visited by the travelling standard Brewer #109 in 2022 and employed in this study to test PHYCS under different conditions.

| Location | Latitude | Longitude |
|---|---|---|
| Hobart, Tasmania | 42.90° S | 147.33° E |
| Norrkoping, Sweden | 58.58° N | 16.15° E |
| Helsinki, Finland | 60.20° N | 24.96° E |
| Sodankyla, Finland | 67.37° N | 26.63° E |
| Broadmeadows, Australia | 37.69° S | 144.95° E |
| Brisbane, Australia | 27.39° S | 153.13° E |
| Toronto, Canada | 43.78° N | 79.47° W |
| Alomar, Norway | 69.23 ° N | 16.01° E |
| Kjeller, Norway | 59.98° N | 11.05° E |

More details are provided in Table 1. The choice of the instruments is based on the fact that each of these stations has at least one double Brewer (MkIII) and at least one single Brewer (MkII, MkIV or MkV) that operated continuously for at least 6 months (to explore the entire range of the solar zenith angles at each location). We analyse the data from all the above instruments in the interval covering the 6 months from 22 December 2019 to 22 June 2020.

Additionally, the data from the travelling standard Brewers #017 and #109, operated by International Ozone Services (IOS), were also analysed. The former instrument allows us to assess the long-term changes in the stray light effect. Indeed, for this spectrophotometer the record from 1993, when the double-monochromator Brewers were first introduced, to 2021 was considered. Additionally, data collected by Brewer #109 in 2022 at several stations were used to test PHYCS under different conditions. In fact, while in recent years Brewer #017 has travelled mostly to central Europe, the IOS standard #109 has visited 9 stations in 2022 in both northern and southern hemispheres with vastly different observing conditions (Table 2). The data from all these sites were processed with a single set of stray light correction factors, to confirm that these coefficients only depend on the instrumental characteristics, not on the location (Sect. 3.2.4).

Finally, to study the stray light effect with radiative transfer simulations (Appendix B2), a set of laser scans measured by the ECCC personnel in the early 2000s (Fig. 1) were employed. These spectra were obtained for several Brewers by scanning the 325.029 nm emission line of a helium-cadmium (HeCd) laser.

## 2.2 Stray light definition, phenomenology and proposed reversal algorithm

When measuring radiation at wavelength $\lambda$ by means of a spectrometer with slit width $\delta\lambda$, the stray light is defined as radiation of wavelengths outside the bandpass $\lambda \pm \delta\lambda$ that reaches the detector.

The presence of stray light is attributed to scatter inside the instrument by impurities and imperfections in the optical elements (Woods et al., 1994). For example, the monochromator in Brewers is routinely opened for maintenance and dust can enter the instrument quite easily. Then, having bright light while measuring solar radiation contributes to both electro- and photophoresis



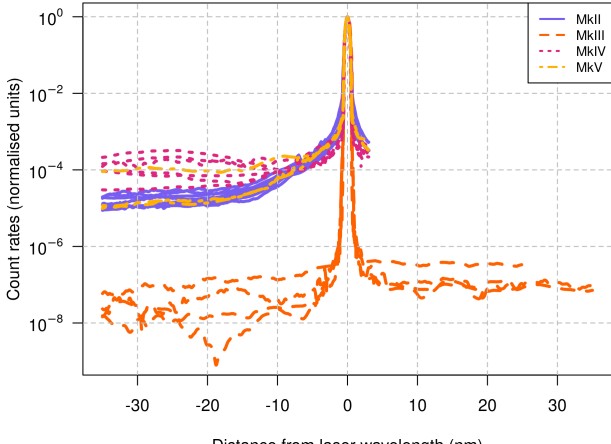

**Figure 1.** Spectra resulting from scanning a 325 nm laser line with different Brewers. The effect of stray light is clearly visible as an increase (in single-monochromator Brewers compared to double-monochromator Brewers) in the count rates at large distances from the excitation wavelength. The spectra were normalised to their respective peak values and coloured based on the Brewer type (MkII: serial numbers #012, #014, #015, #017, #033, #037, #053, #055, #113; MkIII: #021, #085, #107, #111, #128; MkIV: #007, #009, #071, #079, #080, #082, #084, #109; MkV: #029, #039, #042). Scans from single Brewers are incomplete towards larger wavelengths, owing to the reduced spectral range of these instruments, only up to 328 nm.

and dust gets accumulated on optical element surfaces, such as the spectrometer mirror (Kerr, 2002). This ageing leads to an increase in the stray light effect with time. Holographic diffraction gratings are also a very important source of scattering inside
the instrument, depending on their ruling density and line quality. Generally speaking, stray light is larger for gratings with lower line density (Kiedron et al., 2008). Hence, as a rule of thumb, the effect is more pronounced in MkIV Brewers than in MkII Brewers (see Appendix A1).

The difference in stray light among Brewer types is illustrated in Fig. 1, where the responses of several instruments to the quasi-monochromatic radiation emitted by a 325 nm laser are shown. The effect of the resolution of the instrument due to the
finite dimensions of the slits can be noticed as a broadening of a few nanometres around the excitation wavelength ("core" region), while the effect of stray light manifests itself as a non-zero background irradiance at wavelengths far away ("shoulder" and "wing" regions). Although the level of stray light looks very low compared to the signal, even in single-monochromator instruments, its contribution integrated over the whole spectrum has a pronounced effect. The figure confirms that, with the exception of a few instruments, MkIV Brewers show larger stray light than MkII's, and MkIII Brewers reject stray light about
$10^3$ times better than single-monochromator Brewers. It is interesting to note that even Brewers of the same type exhibit some heterogeneity in their stray light levels, likely owing to different mechanical and optical characteristics, their age and state.





To parameterise the stray light and separate it from the "true" (unknown) irradiance, a formal framework is introduced, where the detected count rate, $I_d$, is represented as a sum of the true count rate, $I_t$, and the stray light count rate, $I_s$:

$$I_d(\lambda) = I_t(\lambda) + I_s(\lambda) \tag{1}$$

For a given wavelength, $I_s(\lambda)$ depends on the instrumental characteristics and the intensity and shape of the solar spectrum. Among the wavelengths used in the Brewer observations, slit 5, nominally at 320 nm, is the least affected by the presence of ozone. Hence, the detected count rate at slit 5, $I_{320}$, is used as a proxy for the level of light available for stray light. We can introduce a factor of proportionality $\alpha$ to have

$$I_s(\lambda) = \alpha I_{320} \tag{2}$$

Now, our working hypotheses are that:

1. The stray light contribution $I_s(\lambda)$ (and hence $\alpha$) is only weakly wavelength-dependent in the narrow band used by the Brewer to retrieve ozone and sulphur dioxide (306–320 nm), and it adds light to all slits almost uniformly in the count rate space. The exact values of $I_s$ and $\alpha$ make almost no difference to the bright light at longer wavelengths, but an accurate estimation of $I_s$ at the shortest wavelengths is crucial for ozone and sulphur dioxide retrievals;

2. The coefficient $\alpha$ does not change as a function of the atmospheric and observing conditions. This allows the determination of $\alpha$ through a calibration process (Sect. 2.3) and then use it under all conditions.

To support these hypotheses, stray light in Brewers was simulated numerically (Sect. 2.4 and 3.1).

Based on the assumptions above, the true count rate is simply:

$$I_t(\lambda) = I_d(\lambda) - \alpha I_{320} \tag{3}$$

Note that Eq. (3) is linear (in the "count rate space"). However, as soon as logarithms of the counts are considered while solving the Bouguer-Lambert-Beer's law equation (Bouguer, 1729), the effect becomes highly nonlinear. This is why most previously published research is focused on using a range of complicated nonlinear functions for correcting the ozone values. In contrast, PHYCS addresses the core of the stray light effect and reverses it in the count rates.

From a practical perspective, to mathematically reverse the effect of the stray light in the Brewer data, the detected count 160     rate at each slit is corrected for the dark counts and the dead time, then the stray light contribution is removed using Eq. (3). To be precise, the calculations (Sect. 3.1) show that two stray light coefficients are actually needed for higher accuracy, i.e. $\alpha$ for slits 2–5 and $\beta$ for slit 1. Once the count rates at all slits have been corrected, the ozone and sulphur dioxide, as well as any other product, can be derived using the standard or custom algorithms. This also allows the user to apply PHYCS upstream of other corrections, e.g. to account for changes in the Brewer radiometric sensitivity ("standard lamp" correction).





## 2.3 Determination of stray light coefficients


PHYCS requires a calibration process to determine factors $\alpha$ and $\beta$. Three methods are discussed: the first one uses a reference Brewer that has no measurable stray light or has already been calibrated for the stray light correction algorithm (Sect. 2.3.1); the second method involves absolute calibration from Langley plot observations (Sect. 2.3.2); the third method is based on a statistical analysis of a long-term dataset (Sect. 2.3.3).

In most cases the transfer method will be used, since an absolute calibration can only be performed in a pristine environment with stable ozone, sulphur dioxide and aerosol conditions, and there are only a few locations in the world satisfying this requirement (Zhao et al., 2023).

### 2.3.1 Calibrating using a reference instrument

An iterative calibration process is employed to establish the coefficients $\alpha$ and $\beta$. Since the Brewer algorithm for sulphur
dioxide uses total ozone column, it is important to first establish $\alpha$ and only then proceed with the calibration for $\beta$. Calibrations for $\alpha$ and $\beta$ involve virtually identical steps, so only the process for $\alpha$ is described:

1. The calibration starts with first guesses for $\alpha$ and for the extra-terrestrial coefficient ($ETC_{O_3}$, Appendix A3). $\alpha$ can be taken from model calculations (Sect. 3.1) and the first guess for ETC can be the one in use or calculated from the standard ozone calibration procedure for Brewers as described in Appendix A3. Depending on the observing conditions
during the standard ozone calibration, that first guess is very likely an underestimate of the true ETC that should be used together with the stray light correction;

2. Both $\alpha$ and ETC are varied to have ozone calculations match those from the reference Brewer. This step is similar to the standard Brewer calibration, where only the ETC is varied to match the retrievals from the reference Brewer. Calculations (Sect. 3.1) show that the Brewer retrievals are sensitive to the values of $\alpha$ and ETC at different air mass factor (AMF)
and ozone SCD ranges, which allows the two unknowns to be estimated independently. Any non-linear multi-parameter optimisation algorithm can be employed for the purpose, even a simple trial and error method works well;

3. Unlike the standard ozone calibration, where the AMF is limited to the range of 1.2–3.2, the data from a wider range 1.2–4.5 are used, if available, since the stray light effect is more pronounced at larger slant ozone amounts. The direct-sun data at AMFs larger than 4.5 is affected by the Rayleigh and aerosol scattering in the field of view (Kerr, 2002; Arola
and Koskela, 2004), and will not be considered here.

### 2.3.2 Calibrating using the Langley method

In the rare situation when a Brewer is at a location suitable for absolute calibration (Zhao et al., 2023), the Langley plots can be used to establish both $\alpha$ and the ETC without any additional reference instrument. This method also involves an iterative process, but is simpler in calculations than the previous one since only one parameter, $\alpha$, is varied to have the Langley fit as
close to a straight line as possible. The residual from the fit can be used as the metric, as also done by Vaziri Zanjani et al.



(2019). Then, the ETC is determined as the intercept of the Langley fit in the standard way. This method has, however, significantly higher demand for the quality and the quantity of the data, and for the slant ozone range. First, all known instrumental corrections need to be applied (instrumental temperature, dead time, colour of the neutral density filters, etc.). Second, the physics of stratospheric ozone works against finding a solution in this situation: pristine locations for absolute calibrations are

close to the tropics and the equator because that is where ozone is more stable. Those are also the locations where the total ozone column is lower, making it difficult to collect data at large ozone SCDs, needed for a reliable determination of $\alpha$. The implications of this are addressed in Sect. 4.

### 2.3.3   Monitoring the stability of the stray light coefficients with statistical methods

In places where single Brewers cannot be regularly compared to reference instruments with lower stray light, the user can rely

on statistical methods to monitor the stability of the stray light correction. The method proposed here is based on plotting a large number of individual retrievals (ozone or sulphur dioxide) as deviations from their respective daily medians. If little or no systematic daily variations of ozone and sulphur dioxide are expected, then the differences from the daily medians should be almost randomly distributed around zero. Hence, any clear departure from zero increasing with ozone SCD could be a sign of the stray light effect. Similar techniques were used, for example, by Diémoz et al. (2015) and Stübi et al. (2017) to estimate

the stray light effect on ozone retrievals.

This method is likely not precise enough to provide an absolute estimate of $\alpha$ and $\beta$, but can be very useful to monitor the stray light effect on a continuous basis in absence of a reference Brewer and to identify sudden variations indicating that the instrumental properties have changed and to recover data from a series which was not characterised for the stray light correction.

## 2.4   Radiative transfer simulations

Simulations are helpful to support the hypotheses at the base of our algorithm (Sect. 2.2) and to first test our correction in a controlled setting, e.g., with an a-priori knowledge of the "true" vertical column density. Moreover, they allow us to check the consistency of PHYCS with the existing methods based on radiative transfer calculations (e.g., Kiedron et al., 2008; Karppinen et al., 2015; Vaziri Zanjani et al., 2019). As opposed to other studies, the radiative transfer results were not used to correct the

Brewer count rates, but were only employed to obtain a fist guess of the stray light correction factors $\alpha$ and $\beta$.

Notably, this part of the study consists of three simulations (S1–S3) with different objectives:

–   S1. We first simulate direct-sun Brewer measurements without stray light and perform standard $O_3$ and $SO_2$ retrievals. The solar spectra are calculated analytically by the Bouguer-Lambert-Beer's law. More details about the radiative transfer simulations are provided in Appendix B. It was verified that the retrievals using the standard Brewer algorithm match

the values provided as input to the calculations;





- S2. Stray light was added following the procedure described in Appendix B2. The ETC calibration of this virtual instrument by transfer from the reference (step S1) was also simulated. This reproduces the way network instruments are actually calibrated based on comparison with a travelling standard (Sect. 2.3.1 and Appendix A3);

- S3. $\alpha$ and $\beta$ were estimated and PHYCS was applied to the data obtained at step S2. The corrected $O_3$ and $SO_2$ retrievals are compared to those from the previous steps (and to the a-priori value) to assess the accuracy of the method.

Steps S1–S3 were repeated for several Brewers using their respective experimental characterisation (see Appendix B1). Results are presented in Sect. 3.1.

# 3 Results

The results obtained from the simulations (Sect. 3.1) are provided here to demonstrate the validity of the assumptions and to obtain a first-guess estimate of the stray light correction factors $\alpha$ and $\beta$. Results from real-world observations are given afterwards, in Sect. 3.2.

## 3.1 Evaluation of the method using simulations

The results of the simulations for Brewer #009, this being the instrument with the largest stray light effect among those considered in this study, are presented here. Simulations for the other single-monochromator Brewers listed in Table 1 lead to very similar conclusions and are not shown.

The simulated stray light spectrum is depicted in Fig. 2a as a percentage of the total signal ("true" signal, plus stray light) at each wavelength. It is obvious that the fraction of stray light increases rapidly with ozone SCD. The stray light contribution to the total signal is of the order of 1 % or less for the three longest Brewer wavelengths ($\lambda \geq 313$ nm, i.e. slits 3–5). Based on further calculations, stray light at these wavelengths does not affect the ozone retrieval significantly ($\ll 1$ %) even at the largest slant ozone amounts. Conversely, what affects the ozone and sulphur dioxide retrievals is the stray light at slit 2, reaching 10 % for an ozone SCD of 2000 DU, and at slit 1, reaching 80 %, respectively. As a consequence, in the method it is fundamental to correct the count rates at slits 1 and 2, while the subtraction of the stray light contribution at larger wavelengths (Eq. 3) may be skipped without any loss of accuracy. Furthermore, it should be noticed that all count rates measured through slit 0 are almost entirely due to stray light when the ozone SCD is 2000 DU. Count rates at this wavelength, however, are not used in the current retrieval algorithm.

The same spectra are represented in Fig. 2b after normalisation to the total signal at slit 5, $I_{320}$. These values are equivalent to the correction factors $\alpha$ and $\beta$ introduced in Sect. 2.2 for ozone and sulphur dioxide, respectively, i.e. they represent the fraction of light at 320 nm that should be subtracted from the measurements to obtain the "true" count rates. It can be noticed that this factor changes with wavelength and ozone SCD. However, for slits 1 and 2 (306 and 310 nm) and for large ozone SCDs, where it is most relevant, it remains quite stable, with a value of $\alpha = 2 \times 10^{-3}$ (or 0.20 %) at slit 2 and slightly lower at slit 1, $\beta = 1.7 \times 10^{-3}$ (or 0.17 %). These values refer specifically to Brewer #009 (laser scan, wavelength settings) at a specific



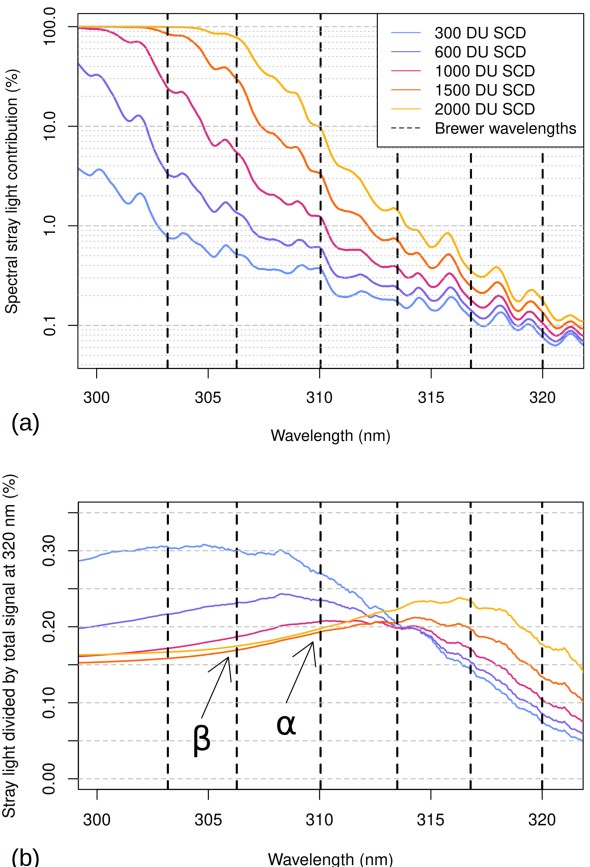

**Figure 2. (a)** Stray light spectrum simulated for different ozone slant column densities (SCDs), divided by the total signal at each wavelength ("true" solar light, plus stray light). Notice that the vertical scale is logarithmic. The high-resolution spectrum used in the calculations is here downscaled to the Brewer resolution (slit 1) for ease of visualisation. **(b)** Same as above, but the stray light spectrum is normalised to the total signal measured at 320 nm. The vertical dashed lines in both figures indicate the centre wavelengths at which the direct solar irradiance is measured. The experimental characterisation of Brewer #009 was used in the calculations.

time of its life (e.g., when the laser scan was performed). However, by examining results from simulations for several Brewers, it was found, as a general rule, that $\beta$ is almost always lower than $\alpha$, and that the changes of these coefficients at large ozone SCDs are small. Variations of the ratio plotted in Fig. 2b at slits 1 and 2 for low ozone SCDs should not be a concern, as the stray light effect is negligible in these conditions (Fig. 2a). Likewise, changes of the ratio at longer wavelengths are irrelevant, as the stray light is not an issue for slits 3–5, based on previous discussion and Fig. 2a.

These preliminary calculations give confidence that the method described in Sect. 2.2 has a solid basis, which is confirmed by the simulations shown in Fig. 3. The figure presents the outcome of the Brewer ozone retrieval performed on synthetic solar spectra. If no stray light is included in the spectra (case S1), then the ozone retrieved with the standard algorithm is very close





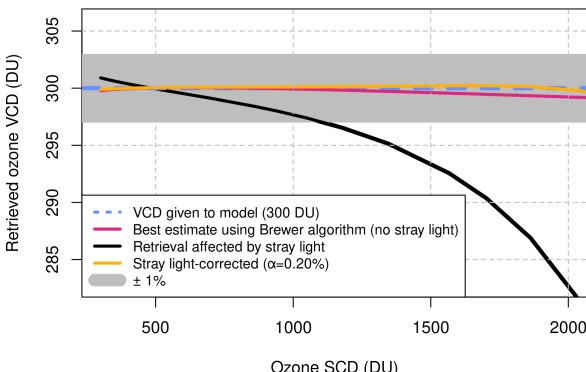

**Figure 3.** Ozone retrieval technique applied to synthetic solar spectra, as a function of the ozone SCD. The true value provided to the model for this case is 300 DU (dashed blue line). The retrieval with a Brewer unaffected by stray light is plotted as a violet line. The stray light effect is simulated for a single-monochromator Brewer (black line, using the characterisation from Brewer #009). Once the correction algorithm is applied to the stray light-affected spectra, the ozone retrievals overlap again to the true value. The grey band represents the $\pm 1$ % interval around the true value.

to the true value given as input to the calculations and well within 1 % (the often-cited goal in the Brewer community is to have the Brewers agree to better than 1 %, which is to say the desired precision in the Brewer network is 1 %). Retrievals from measurements affected by stray light (case S2), however, deviate markedly from the true value, and for Brewer #009 they are off by about 1 % at ozone SCDs of 1000 DU (>5 % at 2000 DU). Depending on the geographic location of the Brewer and the average ozone VCD, a SCD of 1000 DU could be reached for an air mass factor lower or higher than 3. Moreover, the ETC is underestimated: the ozone ETC is 2583 for simulation S1 (no stray light) and 2572 in simulation S2 (affected by stray light). This is due to the fact that the Brewer with stray light (S2) was calibrated by transfer from a virtual reference instrument (S1), and the procedure tries to compensate, on average, the ozone underestimation in the field instrument due to stray light by decreasing the value of the ETC. Quite importantly, the ETC underestimation depends on the chosen (or available) interval of air masses during the calibration procedure. This topic will be addressed in the Discussion (Sect. 4).

When a stray light correction with the proper $\alpha$ coefficient is applied (simulation S3), however, the results are significantly improved and the retrievals are brought back to about the true value even for ozone SCDs of 2000 DU. The improvement is impressive, especially considering that the correction is based on a single free parameter. The ozone ETC, in simulation S3, perfectly agrees with the one obtained without stray light in the model (2583).

An important remark is that the $\alpha$ parameter only depends on the instrument, not on the environmental conditions. Indeed, PHYCS was tested with different values for the ozone VCD (250 to 500 DU, reproducing tropical to polar atmospheres), atmospheric pressure, aerosol amount and Ångström exponent, and the value of $\alpha$ providing the best correction was still the same (not shown, as experimental evidence of this property will be given in the next sections).




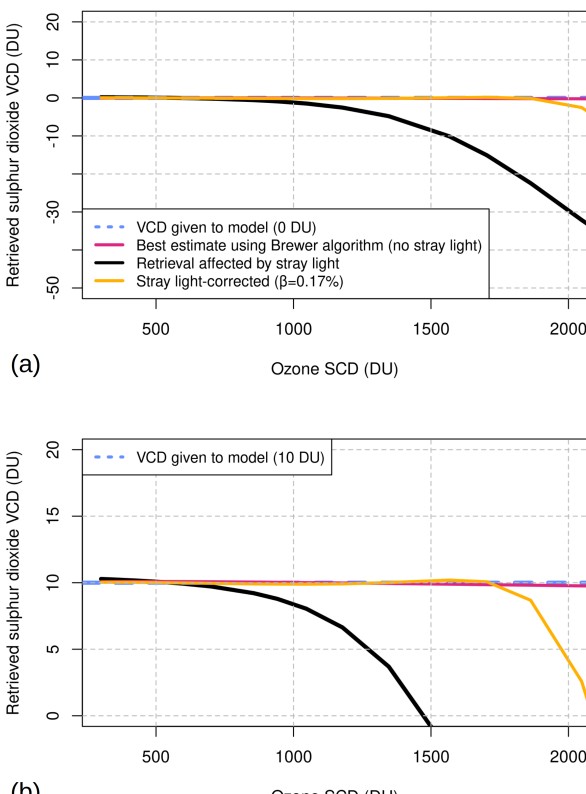

**Figure 4.** Sulphur dioxide retrievals from the synthetic solar spectra, as a function of the ozone SCD. The true value used in the calculations is **(a)** 0 DU (dashed blue line) and **(b)** 10 DU. Notice that the vertical scales in the two subfigures are different.

The same procedure was applied to sulphur dioxide retrievals. In the first case (Fig. 4a), no $SO_2$ was included in the calculations. This scenario represents a typical situation for most Brewer stations, located far from strong sources of sulphur dioxide. The results show that, in pristine conditions, stray light leads to negative concentrations of $SO_2$, as frequently observed in the experimental data (Hrabčák, 2018). As in the case for ozone, a proper choice of the $\beta$ coefficient allows the $SO_2$ Brewer retrievals to be effectively corrected. In the second case (Fig. 4b), a very high sulphur dioxide VCD (10 DU) was set, likely representative of measurements close to a volcanic plume (e.g., Zerefos et al., 2017). The figure highlights that the same $\beta$ factor previously determined for pristine conditions still stands in a scenario with high $SO_2$ concentrations and that PHYCS allows accurate sulphur dioxide retrievals up to rather large air masses (or ozone SCDs). As for ozone simulations, $ETC_{SO_2}$ is underestimated when stray light is included the calculations (2243, case S2), while the values are about the same in cases S1 (2279, no stray light) and S3 (2278, corrected with PHYCS). However, as explained in Sect. 3.1, this has little practical implication, since sulphur dioxide only weakly depends on the extra-terrestrial coefficient.



Using the calculations, additional hypotheses can be tested. First of all, it can be shown that the algorithm only works if the
parameter $\alpha$, i.e. the percentage of the count rates deemed to be representative of the stray light, is calculated with reference
to the total signal through slit 5. If $\alpha$ is calculated, e.g., with reference to slit 3 (Fig. S2), then the performances deteriorate
significantly. This is due to the fact that wavelengths shorter than 320 nm are strongly affected by ozone absorption, thus they
are not a suitable reference to calculate the stray light contribution at slit 2.

A second numerical experiment can be performed to assess the effect of the uncertainty of the stray light correction factors.
To this end, the changes observed in the ozone retrievals when using slightly larger ($\alpha_+$ = 0.3 %) or slightly lower ($\alpha_-$ =
0.1 %) correction factors than the optimal one ($\alpha$ = 0.2 %, for Brewer #009) were calculated. As evident from Fig. S3a for
ozone, variations in $\alpha$ affect the retrievals at large ozone SCDs. It is interesting to note that, conversely, errors in the ETC
determination ($\pm 10$ units in the example, a slightly larger range than expected in ideal conditions) mostly affect the ozone
retrievals at small air masses (Fig. S3b). Hence, the orthogonality between $\alpha$ and ETC allows the accurate assessment of the
two unknowns independently without significant cross-talks. Figure S3b also shows that the Bouguer-Lambert-Beer law for
the Brewer has reversed sign (see Appendix A2) and that increases to ETC lead to apparent decreases in ozone.

The same exercise was repeated for sulphur dioxide retrievals, by changing either $\alpha$ or $\beta$. Figure S4a reveals that the $\alpha$
parameter has an influence on the $SO_2$ retrievals, as the optical depth of ozone must be subtracted to the total optical depth
at 306 nm from which sulphur dioxide is calculated. Therefore, $SO_2$ calculations are very sensitive to the accuracy of ozone
(about -0.5 DU change in $SO_2$ for every +1 DU change in ozone), which explains why it is necessary to first obtain the correct
estimate of $\alpha$ and then for $\beta$. Figure S4b, instead, shows the effect of small variations in $\beta$ on sulphur dioxide retrievals. The
sensitivity to this parameter is also high, and for Brewer #009 variations of sulphur dioxide VCD of $\pm 5$ DU at 1800 DU ozone
SCD can originate from changes of $\pm 2 \times 10^{-4}$ in $\beta$. Finally, Fig. S4c highlights that the dependence of the retrieved sulphur
dioxide VCDs to $ETC_{SO_2}$ is very low (about 0.34 DU change in $SO_2$ for every 10 units in ETC at air mass factor 1).

Finally, a third test was conducted to explore the feasibility of determining the stray light correction coefficients from a
Langley plot (Sect. 2.3.2), i.e. without the need of any reference instrument for the transfer of $\alpha$ and $\beta$. Figure S5 shows a
simulated Langley plot for ozone ($R_6$ linear combination) with the single-monochromator Brewer #009. Some features are
obvious: first of all, the results show that Langley plots with measurements affected by stray light make little sense, as the
linearity of the relation between $R_6$ and air mass is strongly degraded; second, the method allows an effective correction to
the Langley plot, and this fact could be employed to retrieve the correction factors by choosing the $\alpha$ and $\beta$ values leading to
the straightest line, provided that ozone is very stable. It should be noted, however, that the feasibility of this method depends
not only on the available air mass range, but also on the ozone VCD, as the stray light effect manifests itself at large SCDs (in
Fig. S5, the ozone VCD used for the calculations is 300 DU). As mentioned in Sect. 2.3.2, the best sites for Langley plots, i.e.
where the ozone VCD is assumed to be the most stable, are located in regions of the world where the ozone layer is thinner,
hence the stray light effect might be less evident there due to the lower SCDs. Finally, the figure also shows that the residuals
after the stray light correction are even lower than the residuals without stray light in the model. This likely means that the
stray light correction also compensates, in small part, for inaccuracies related to the retrieval algorithm itself rather than to the
stray light effect. A slight tendency to underestimate ozone at large SCDs by the Brewer algorithm was also found in previous





studies (Kiedron et al., 2008). Calculation of instrument-specific weightings in $R_6$ could help in removing this small residual
effect (Savastiouk and McElroy, 2005).

## 3.2   Application to experimental data

### 3.2.1   Ozone and sulphur dioxide observations at three Brewer stations

Figure 5 depicts a comparison of the ozone and sulphur dioxide retrievals before and after PHYCS implementation on real data.
As described in Sect. 2.1, three locations characterised by very different atmospheric conditions were chosen to test PHYCS
in several scenarios.

To assess the performances of the stray light correction on ozone (Figs.5a, c and e), the retrievals from three single Brewers
are compared with the retrievals from co-located MkIII Brewers used as references. Measurement pairs within 5 minutes from
single and double Brewers were considered. Uncorrected data show systematic underestimations of about -2 to -3 % at 1600
DU ozone SCD at all sites due to stray light. Owing to higher ozone SCDs reached during the year at the Alert site, compared to
the other locations, the deviations of the MkII retrievals from the MkIII are as big as -6 % at 2100 DU ozone SCD. However, the
new algorithm works well under all atmospheric conditions and, after application of PHYCS, the corrected values are virtually
the same as those retrieved from the double-monochromator spectrophotometers. Indeed, most of the corrected retrievals fall
within ±1 % with respect to the double Brewers, even at very large ozone SCDs. At Mauna Loa Observatory, the agreement
is better than 1 % up to SCD ozone of 1900 DU, which corresponds to approximately an air mass factor of 5.5. Field-of-view
stray light and very dim light contribute to high noise level at large solar zenith angles. Points collected at very low SCDs show
high retrieval noise, both in the corrected and uncorrected data, owing to minimal absorption by ozone.

The stray light correction factors employed in PHYCS were 0.40 % (MkIV Brewer #009), 0.33 % (MkII Brewer #008)
and 0.29 % (MkII Brewer #029), which agrees with the stronger stray light effect expected in MkIV Brewers compared to
MkII's. Table 3 also reports the factors for the two travelling standards (#017 and #109), with comparable values. It might be
noted that the experimental correction factor for Brewer #009 is of the same order of magnitude as the one determined with
the model in Sect. 3.1, but larger. This can be due to the simplified assumptions in the calculations, slight discrepancies in the
configuration used for the retrieval and, above all, the ageing of the instrument between the time the laser scan was performed
(around year 2000) and the time when the data shown in the figure were collected (2019–2020). Indeed, stray light has likely
increased in the course of twenty years, as shown in Sect. 3.2.5 for Brewer #017. Regarding the ETCs, Table 3 shows that
the true extra-terrestrial coefficients are always significantly larger than those originally used to provide (stray light-affected)
retrievals matching the reference measurements. The magnitude of these differences depends largely on the minimum SCD
used for the original calibrations.

The results for sulphur dioxide VCDs (Figs.5b, d and f) similarly show that all corrected data fall within ±1 DU when it
is expected to be zero (no strong $SO_2$ sources close to the stations) and that, after applying PHYCS, the deviation from zero
has a similar distribution in both the single and the double monochromator. As predicted by the calculations, the coefficients
$\beta$ used for $SO_2$ retrievals for the three Brewers are lower than the $\alpha$'s and they comply with the ordering expected by Brewer





**Figure 5.** Comparison among retrievals affected by stray light and corrected using PHYCS. **(a, c, e)** Comparison of ozone VCDs retrieved by three single Brewers at different locations and three co-located double Brewers. The data shown were collected in the period December 2019 to 22 June 2020. **(b, d, f)** Absolute values of sulphur dioxide VCD retrieved by the three single Brewers, before and after stray light reversal. Points represent single measurements pairs (ozone) or individual measurements (sulphur dioxide). The circles and the error bars indicate the average and the standard deviation of data binned at 100 DU ozone SCD intervals.



**Table 3.** Configuration of the Brewers used in the study before and after implementing PHYCS.

| Brewer | #009 (MkIV) | #008 (MkII) | #029 (MkII) | #017 (MkII)[a] | #109 (MkIV) |
|---|---|---|---|---|---|
| MkIII reference | #119 | #145 | #246 | #145 | #145 |
| $\alpha$ (%) | 0.40 | 0.33 | 0.23 | 0.19 | 0.40 |
| $\beta$ (%) | 0.39 | 0.12 | 0.11 | 0.16 | 0.28 |
| $\text{ETC}_{O_3}^{\text{true}}$ | 2735 | 3180 | 3180 | 2855 | 3070 |
| $\text{ETC}_{O_3}^{\text{orig}}$ | 2722 | 3170 | 3150 | 2840 | 3055 |
| $\Delta\text{ETC}_{O_3}$ | +13 | +10 | +30 | +15 | +15 |
| min AMF | 1.00 | 1.07 | 1.93 | 1.07 | 1.07 |

[a] Data presented in this table for Brewer #017 correspond to the last row in Table 4.

type (MkIV v. MkII). The extra-terrestrial coefficients for $SO_2$ are not reported in Table 3, as sulphur dioxide retrievals show very little sensitivity to variations in ETC, and no change in ETC after applying PHYCS was deemed necessary.

Depending on the atmospheric conditions, sulphur dioxide retrievals can be more or less noisy at extreme ozone SCDs as seen in these figures. This is because of the very rapid dimming of the shortest wavelength (306 nm) that is only used for $SO_2$ retrievals. Filtering data with a minimum required count rate at this wavelength can reduce the number of questionable points in the final product. All points where the detected count rate is lower than the estimated stray light contribution should be eliminated.

### 3.2.2 Sensitivity to stray light coefficients using observations

As done in Sect. 3.1 using simulations, tests were performed on the experimental observations to assess the sensitivity of the stray light correction to $\alpha$ and $\beta$. Data from Brewer #009 were employed for that purpose. The $\alpha$ factor was changed to 0.50 % and 0.30 % (its optimal experimental value being 0.4 %), and the maximum changes were of the order of about 1 % for ozone (at 1700 DU ozone SCD) and 1–2 DU for $SO_2$ (Fig. 6). The same variation in $SO_2$ was found for much smaller changes of $\beta$, its optimum value of 0.39 % for Brewer #009 being changed to 0.41 % and 0.37 %. The results of this test with real observations are similar in magnitude to those from the model, although the experimental data in Mauna Loa do not reach the same large amounts of ozone SCD as in the simulations. A slightly lower sensitivity can be noticed in the real data, which is compatible with the larger correction factors needed in the observations with respect to the calculations.

If it is assumed that absolute variations of $\alpha$ of $\pm 1 \times 10^{-3}$ occur on average in about 10 years (Sect. 3.2.5), we can conclude that in the same lapse of time the ozone retrievals corrected for stray light will still be within 1 % with respect to a MkIII Brewer up to approximately 1700 DU SCD. This same variation in the ozone correction factor affects $SO_2$ calculations only slightly. A similar period of time is needed to see variations of 1–2 DU in sulphur dioxide due to drifts in $\beta$.



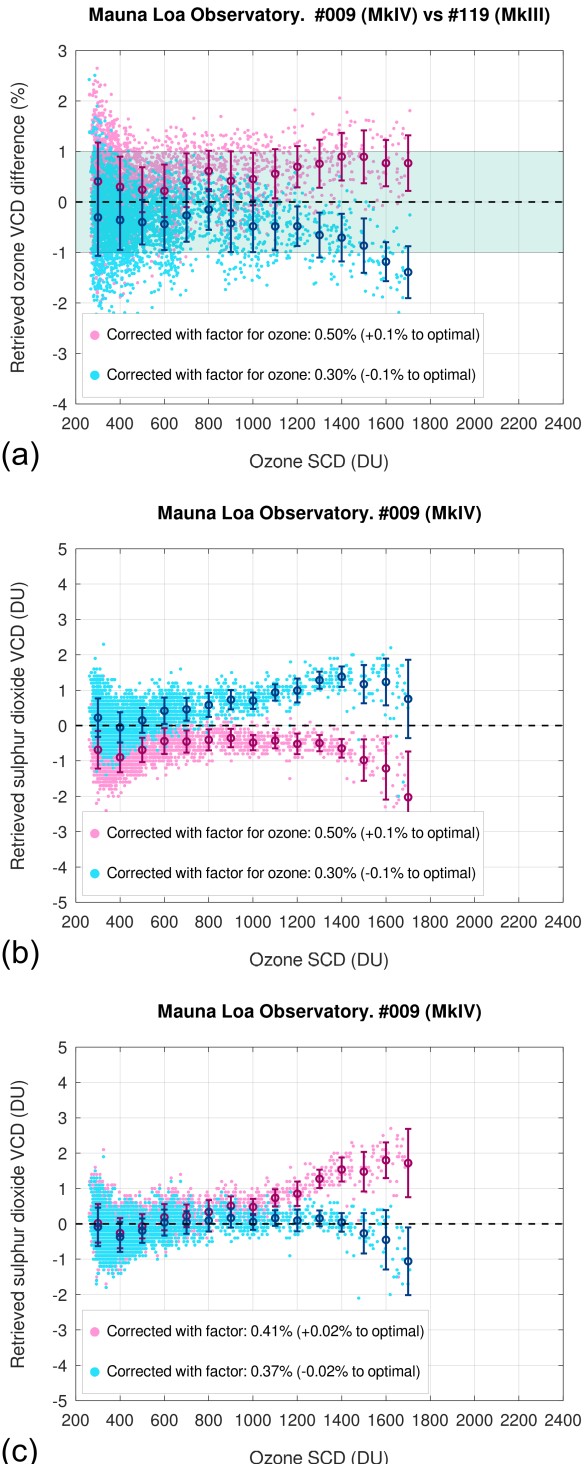

**Figure 6.** Sensitivity test on real observations from Brewer #009 to assess the variation of **(a)** the ozone retrievals to small changes in $\alpha$ (an offset of $1 \times 10^{-3}$ was added and subtracted to the optimal value of 0.40 %), **(b)** the sulphur dioxide retrievals to small changes in $\alpha$ (this latter was again offset by $\pm 1 \times 10^{-3}$) and **(c)** the sulphur dioxide retrievals to small changes in $\beta$ (offset of $\pm 2 \times 10^{-4}$ about the optimal value of 0.39 %). Notice the different y scales.





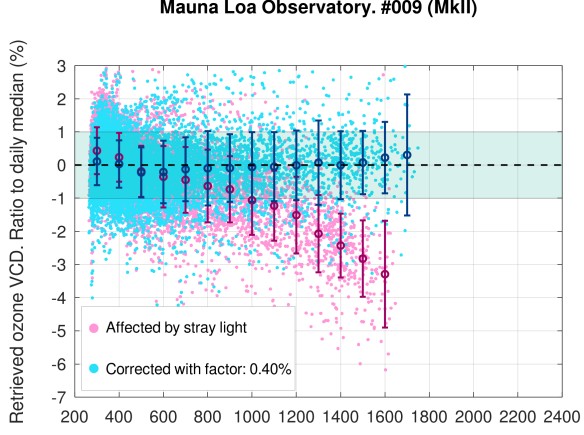

**Figure 7.** Demonstration of the statistical method for checking the stability of the stray light coefficients (Sect. 2.3.3). The vertical axis represents the percentage deviation of individual ozone retrievals from their respective daily median VCD. Data are binned every 100 DU SCD.

### 3.2.3 Testing the statistical method

Figure 7 shows an application of the statistical method described in Sect. 2.3.3 with observational data from Brewer #009. For each ozone retrieval, the percentage deviation from the respective daily median VCD is plotted, then data were binned.
Bins with only a few points ($< 10$) were not plotted due to insufficient statistics. The figure shows similar patterns as the comparison with the co-located MkIII (Fig. 5a), with only a slight increase of statistical noise owing to the replacement of actual measurements with daily medians. The results demonstrate that the method is a useful tool for monitoring the validity of the stray light correction factors for those single Brewers that cannot be regularly compared to a double Brewer.

### 3.2.4 Effect of different atmospheric conditions

To study the effect of different atmospheric conditions, all data from the 9 locations visited by Brewer #109 in 2022 where merged together, and PHYCS was applied using fixed coefficients to the whole data set. Then, the statistical method described in Sect. 2.3.3 was used to check the performances of the stray light correction. Despite some noise, it is clear from Fig. 8 that PHYCS worked well at all sites, and that the $\alpha$ coefficient did not vary markedly with the location. The same is true for sulphur dioxide: the absolute values of the corrected $SO_2$ retrievals, always close to zero, show that a unique $\beta$ coefficient provided
an accurate correction for stray light at all sites. The results show that $\alpha$ and $\beta$ mainly depend on the characteristics of the instrument.





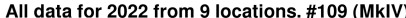

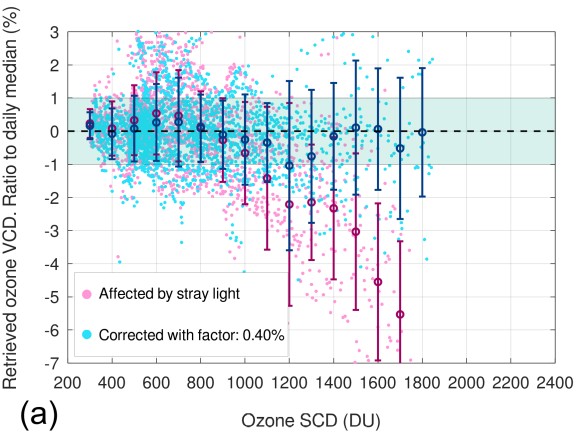

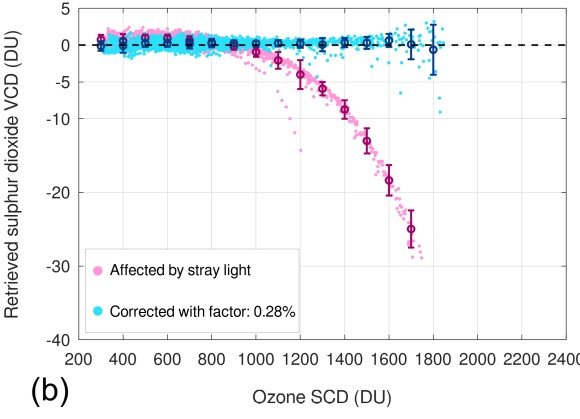

**Figure 8.** Test to assess the effect of location on the stray light correction. All data from 9 locations visited by Brewer #109 in 2022 where merged together. **(a)** Results from the statistical method. **(b)** Retrieved sulphur dioxide VCD.

### 3.2.5 Changes in stray light with time

Data from MkII Brewer #017, together with data from various MkIII Brewers over a number of years, were used to establish historical values for the stray light correction factors. The series were examined in search of retrievals of ozone and sulphur dioxide from the two types of instruments. For this to be done, a large enough range of ozone SCDs is needed, which was not always available in the early years due to the measurement schedules. Late spring was considered the best period for measurements in Toronto, as ozone VCDs are still large and the range of air mass factors is rather wide. A first comparison with data collected in 1993, when MkIII Brewers became operational, was made. Other comparisons at regular intervals were performed so that increases of about 0.1 % in $\alpha$ could be tracked. Table 4 presents the results for some of the comparisons.





**Table 4.** Stray light correction factors for Brewer #017 from 1993 to 2019.

| Year | Reference Brewer | $\alpha$ (%) | $\beta$ (%) |
|---|---|---|---|
| Prior to 1993 (best estimate) | — | 0.19 | 0.16 |
| 1993 | #085 | 0.20 | 0.17 |
| 1996 | #085 | 0.30 | 0.20 |
| 2008 | #145 | 0.40 | 0.22 |
| 2019[a] | #145 | 0.19 | 0.16 |

[a] A new spherical mirror was installed in Brewer #017 in 2019.

It is clear that the stray light increased over the life of the instrument, $\alpha$ changing from 0.20 % to 0.40 % and $\beta$ from 0.17 % to 0.22 %. In 2019, the spherical mirror in the monochromator of Brewer #017 was replaced with a brand new one. The instrumental modification, evidently, had a significant impact on the stray light correction factors, bringing them essentially to the level of the first available comparison with a double Brewer. Not shown here, similar improvements in the correction factor values were seen in all 3 Brewers that have undergone the main mirror replacement.

This has three implications in our opinion:

– First, this suggests that the spherical mirror is a major contributor to the internal stray light and the main degrading component in the optics. Dust accumulation is likely to occur there due to both electro- and photo-phoresis effects while the optics is exposed to bright light;

– Second, it is very likely that the correction factors in the period prior to the first comparison with the double were close to the values in 1993 and 2019 when the spherical mirror was in similar conditions. Thus, our best estimate for the coefficients of Brewer #017 prior to 1993 corresponds to their values in 2019. Even if the former ones were slightly lower, this will not affect the data much and reprocessing of the entire record of Brewer #017 from 1984 to present using PHYCS is now possible;

– This observation suggests that visual inspection of the mirror and its replacement or resurfacing would be a useful maintenance consideration either on the basis of inspection or simply an action taken after a considered period of time, especially if the Brewers are expected to continue operating for a long time. It should be noted that resurfacing the mirror would more likely guarantee the instrument would remain in its former alignment (dispersion, slit width).

### 3.2.6 Uncertainty estimation

Thanks to PHYCS, accurate retrievals from single Brewers are now possible even for large ozone SCDs. Without a proper correction, measurements in such conditions would have normally been filtered out from the analysis by the operator owing to the dominant effect of stray light. The correction for this systematic effect with PHYCS and the resulting extension of the measurement range come at a price, i.e. a slight increase in the overall retrieval uncertainty due to the introduction of the



correction itself (BIPM et al., 2010). Here we assess the contribution of the stray light correction to the retrieval uncertainty only for ozone, as sulphur dioxide retrievals are impacted by other and more important sources of uncertainty (Fioletov et al., 2016; Zerefos et al., 2017).

The best way to assess the contribution of the stray light correction to the ozone retrieval uncertainty is to start from Eq. (2), which is the mathematical representation of PHYCS, and to consider how the uncertainty of the different terms propagates to ozone. Three sources of uncertainty can be identified:

1. Uncertainty in the counts measured at 320 nm, $I_{320}$. This component is dominated by Poisson noise in photocounting
detectors and is expected to increase at very large SCDs. It can also depend on the sensitivity of the particular Brewer and on the dark counts. Kerr (2002) and Gröbner and Meleti (2004) state that the Brewer direct irradiance measurements can reach a precision of 0.1%, limited by Poisson noise. Additional calculations show that at slit 5 Poisson noise is $< 1$ % even for solar zenith angles slightly larger than $80°$, where uncertainties from other parameters become more important. Therefore, even if an extreme value of 1% is considered for random noise in $I_{320}$, this would trigger the same
effect on the stray light estimate $I_s$ as a change of $\pm(0.01 \times \alpha)$ in $\alpha$, due to their relation in Eq. (2). Based on the results presented herein, $\alpha$ is always lower than about 0.65 %, and simulations show that Poisson noise in $I_{320}$ would result in uncertainties $< 0.2$ % in ozone (through the stray light correction) at 2000 DU SCD;

2. Uncertainty in the correction coefficient $\alpha$. This component, in turn, is the sum of:

     a. The uncertainty propagated from the reference instrument employed for the calibration transfer to the field instru-
ment, in case the former has already been calibrated and corrected for stray light. If the reference did not need any correction, this term is not considered;

     b. The accuracy and repeatability of the $\alpha$ estimate, depending on the available experimental data and on the method used by the operator to find the correction coefficient leading to the best match with the reference. Some tests were made by limiting the available data to $< 1000$ DU SCD and the effect when the $\alpha$ factor changed by $\pm 1 \times 10^{-4}$,
however small, was still visible. Hence, the stray light correction coefficient can be determined to within $\pm 1 \times 10^{-4}$. Based on radiative transfer calculations, this translates into the retrieved ozone as a change in $\pm 0.2$ % at 1800 DU SCD and $\pm 0.3$ % at 2000 DU SCD;

     c. Drifts in $\alpha$ with time (Sect. 3.2.5), which includes both an increase of stray light due to ageing of the instrument and variations in the spectral sensitivity of the Brewer, altering the solar spectrum "seen" by the instrument. For
a well-maintained Brewer, we assume that $\alpha$ does not drift by more than $\pm 1 \times 10^{-4}$ before the next calibration, hence the same results as in case 2b can be used;

3. The uncertainty of the mathematical model itself, which depends on the reliability of the underlying assumptions. The deviations from these assumptions can vary among instruments, e.g. based on the laser response shown in Fig. 1, there-fore this factor is considered to be uncorrelated among Brewers. A rough estimate of the model uncertainty can be
obtained from Figs. 5a, c and e, and examining the deviations of the bin averages from the 0 % reference line. As they





**Table 5.** Maximum (at 2000 DU ozone SCD) contribution of the stray light correction to the uncertainty of ozone retrievals for primary and secondary instruments.

| Contribution | Primary | Secondary |
|---|---|---|
| Poisson noise in $I_{320}$ | 0.2 % | 0.2 % |
| Transfer of $\alpha$ | — | 0.5 % [a] |
| Data sampling | 0.3 % | 0.3 % |
| Drifts in $\alpha$ | 0.3 % | 0.3 % |
| Mathematical model | 0.3 % | 0.3 % |
| Total (1-$\sigma$) | 0.6 % | 0.8 % |

[a] Poisson noise in the primary instrument is not propagated to the secondary instrument since the number of simultaneous measurements is assumed to be large.

are always within $\pm 0.5$ %, the resulting uncertainty provided by this factor is about $0.5\ \%/\sqrt{3} = 0.3$ % if it is assumed that errors from the simplified model are uniformly distributed.

Each successive level in the transfer of the stray light calibration leads to slightly larger uncertainties, in that the uncertainty of the reference instrument data are propagated to the secondary instrument. Assuming that the number of simultaneous measurements collected during the transfer is large, we can neglect Poisson noise from the primary instrument in the uncertainty of the secondary instrument. Table 5 summarises the factors considered here and the total contribution of the stray light correction to the uncertainty of individual ozone retrievals. In the extreme case of 2000 DU ozone SCD, this contribution is 0.6 % for primary instruments (e.g., calibrated from an instrument without stray light) and 0.8 % for secondary instruments (i.e., calibrated with an instrument already characterised for the stray light effect). These are most likely very conservative (worst case scenario) estimates.

### 3.2.7 MkIII Brewers and stray light

The available laser scans (Fig. 1) show that the expected level of stray light in double-monochromator Brewers is very low, but it is not zero. To our knowledge, no study of stray light in MkIII Brewers exists. The spectral sensitivity curve in double monochromators is usually a monotonically and rather steeply increasing function of wavelength. This can magnify the potential effect of stray light, since the shortest wavelengths are relatively dimmer than those in the single-monochromator instruments given the same light level at the longest wavelength. In addition, the absence of any cut-off filters in the MkIII Brewers can increase the available long-wavelength radiation scattered inside the instrument to reach the detector.

One way to assess whether MkIII Brewers have effects from stray light is to look at the sulphur dioxide retrievals at large ozone SCDs. In some cases, where observations are done at ozone SCDs larger than 1600 DU, the sulphur dioxide retrieved values show significant underestimations with respect to the expected value 0 DU (Fig. S6). Applying our algorithm to the data with no correction for ozone and only a very small correction factor for $SO_2$ completely compensates the effect and the





value of 0 DU is retrieved over the whole ozone SCD range as expected. The correction factor $\beta$ needed for MkIII is an order of magnitude lower than that for single-monochromator instruments. While this simple test is not conclusive, it may suggest that a further investigation of stray light in MkIII Brewers might be useful.

Statistical analysis of ozone using differences of individual values and the daily medians (Sects. 2.3.3 and 3.2.3) show no significant stray light effect in MkIII instruments, but this method is not precise enough as discussed earlier. One possible way to investigate the stray light effect in a particular MkIII on ozone calculations is to have a number of MkIII Brewers at a location where large ozone SCDs can be reliably measured. Then, a comparison of data retrieved with each Brewer can show whether any instruments deviate from the ensemble. Importantly, the more MkIII Brewers that could participate in such a campaign,

the more reliable the results would be. Whatever level of stray light might be in the MkIII's, it is not expected to be largely different among the Brewers. This means that the retrieved ozone data will be very similar among the participating instruments and it will take a large volume of data to definitely conclude whether any of the instruments have a significant stray light effect in ozone.

## 4   Discussion

The discussion now focuses on the following question: why a Brewer user should implement the stray light correction described in this study? There follows a list of some reasons:

1. First of all, PHYCS allows access to a larger range of ozone SCDs and air mass by the Brewer. The common practice of cutting the measurements at large solar zenith angles potentially introduces some biases and might not solve the stray light issue for the most affected Brewers. Also, measurements at different air masses, both small and large, may be of

interest, for example, to studies on ozone trends in different layers of the atmosphere (Fountoulakis et al., 2021);

2. Partially connected to the previous point, a further issue triggered by presence of stray light, as illustrated in Sects. 3.1 and 3.2.1, is the underestimation of the extra-terrestrial coefficient resulting in a calibration transfer. To further complicate things, the magnitude of this underestimation depends on the range of air masses available during a specific calibration campaign. Generally speaking, an erroneous ETC is not a real problem as long as measurements are always performed

at ozone SCDs comparable to those when the ETC was determined. Indeed, the aim of a calibration transfer is to make ozone retrievals from the field instrument to agree with the reference. However, air mass changes through the day and the seasons, which can induce biases with cycles on different temporal scales. Additionally, if a Brewer calibrated in such a way is transported to locations at different latitudes, and operates with very different ozone SCDs, its calibration might not be suitable for the new site. A not dissimilar case is when Brewers normally operating at high latitudes are calibrated

at low latitudes during intercomparison exercises. As during the transfer the ETC is normally calculated to match the ozone VCD from the reference at low airmasses, the calibration might not be accurate when the instrument returns to its original location and operates at larger SCDs.



A few word, but an important addition to the discussion. On a historical note, it should be noted that for specific Brewers strongly affected by stray light, the only way in the past to bring their ozone retrievals close to the reference Brewer during calibration was not only to adjust the ETC, but also to let the ozone absorption coefficient vary (the so-called "two-point calibration"). Therefore, the value of the latter could be different from that calculated analytically based on the laboratory cross-sections. Although the agreement to the reference was all artificial, as both ETC and absorption coefficient were wrong, and still limited to approximately 1000 DU ozone SCD, any other combination would lead to erroneous values of ozone at almost any SCD. This practice should be discontinued in favour of using PHYCS;

3. As quantitatively discovered for the first time, the magnitude of the stray light effect can change during the lifetime of a Brewer. When long-term trends are to be studied, this factor should be considered in order not to introduce fictitious trends, moreover depending on the ozone SCD. On a relative scale, as also mentioned by Kiedron et al. (2008), the increase of stray light effect with time might have larger relative effect for MkIII Brewers. This directly addresses the issue of whether long-term ozone trends have been miscalculated when single Brewers have been replaced by double Brewers, particularly at high-latitude stations;

4. Finally, as PHYCS is implemented at a very early stage of the Brewer data reduction, the same method could be easily adapted to other measurement geometries and techniques in addition to the direct-sun method (e.g., zenith-sky, Umkehr, etc.). Although further advances are needed, PHYCS could contribute to drastically improve the quality of $SO_2$ retrievals with Brewers.

In summary, we believe this algorithm, once implemented, will contribute to better data quality and consistency across:

– zenith angles,

– ozone SCD,

– locations,

– calibration conditions,

– age of the Brewer,

– Brewer types,

– the whole Brewer network.

The simplicity of the algorithm allows it to be easily implemented in the data processing, either by the PIs for individual instruments or by central authorities like the World Ozone and UV Data Centre (WOUDC) and/or the European Brewer network (EUBREWNET). To this aim, we have developed a dedicated multi-platform software package implementing PHYCS and we have made it available to the Brewer users. Also, the most popular Brewer ozone data processing software, O3Brewer (Stanek, 2023), has now been modified to include PHYCS. Past calibrations can be used to re-evaluate the series with no need



for additional data. Moreover, the analysis shows that the correction factors for the algorithm change slowly. As an example, for Brewer #017 a change in the ozone factor of 0.1 % and in the sulphur dioxide factor of 0.02 % was found approximately
every 10 years. This suggests that a usual calibration frequency for the Brewers of 1–3 years is sufficient to keep the stray light correction factors up to date with no effect on the data.

As a simpler alternative for the Brewers that have no calibration for stray light correction, the operator can use some typical values from this study. Indeed, to establish a range for the correction factors, we analysed over 20 Brewers, both MkII and MkIV. While the upper limit was actually similar in both types, the minimum values were significantly different. The range of
$\alpha$ for the MkII was 0.19–0.6 and for the MkIV it was 0.4–0.65. The lowest ozone correction factors are likely an underestimate, but using them will definitely extend the range of ozone SCDs to make the data consistent with stray light-free instruments. For the ETC adjustment, the noon values for ozone VCD can be replicated for the data with and without the stray light correction as the best available estimate in these cases.

In conclusion, two final remarks. First, in this work ozone absorption coefficients based on the Bass and Paur (1985) cross-
sections were used. Since this is a simple scaling factor, it has no effect on the corrections for stray light. This also means that if the Brewer community switches to other cross-sections in the future, the stray light correction parameters will not change. Second, stray light is not expected to affect Brewer retrievals obtained in the visible range, such as nitrogen dioxide from MkIV instruments (Diémoz et al., 2021) or AOD in the interval 425–453 nm (Diémoz et al., 2016), since the spectral gradient of the solar irradiance is less pronounced in this band.

**5   Conclusions**

A new method, the PHYSically-based Correction for Stray light (PHYCS), was developed to correct for stray light in Brewer spectrophotometers and its implementation as a software package is now available to the user community. PHYCS corrects the count rates collected by the instrument at the very beginning of the standard data reduction, therefore it can be used upstream of any additional software normally employed by the operator and does not require further modifications of the processing chain.
Radiative transfer calculations were performed to verify the assumptions at the basis of PHYCS and to provide a first guess of the correction parameters. The effectiveness of the method was then showcased using real-world data from several Brewers. Once applied, PHYCS allows the retrieval of ozone and sulphur dioxide from single-monochromator instruments within $\pm 1$ % and $\pm 1$ DU, respectively, to those from double-monochromator Brewers even at large ozone slant column densities. Only one free parameter for ozone and one for sulphur dioxide are needed for the method, and they can be effortlessly transferred during
calibrations or retrieved by Langley plot analysis. An additional method is proposed to monitor the stability of the stray light effect with time. The uncertainty brought by the stray light correction to the overall ozone uncertainty is estimated to be well below 1 % even for ozone SCDs of 2000 DU.

In addition to the aforementioned findings, this research has yielded further outcomes that contribute to understanding the ozone and sulphur dioxide measurement. For the first time, using a data set longer than 30 years, this analysis has provided
evidence that stray light can increase during the lifetime of an instrument, which can propagate in the calculation of the trends



as a function of the ozone slant path. On another topic, although more investigations are required, the analysis has raised the possibility that even MkIII Brewers could be marginally affected by stray light, especially at the shortest wavelength used for sulphur dioxide retrievals.

PHYCS is already being tested at single stations, such as the Boundary Layer Air Quality-Analysis Using Network of Instruments (BAQUNIN) supersite promoted by the European Space Agency (Iannarelli et al., 2022), and can be readily implemented by central authorities like the World Ozone and Ultraviolet Data Center (WOUDC) or the European Brewer Network (EUBREWNET). In the near future, it is planned to test the algorithm for other observation geometries, different ozone retrievals techniques (e.g., Umkehr) and other products such as the aerosol optical depth (López-Solano et al., 2018). Moreover, PHYCS unveils new potential applications to accurate retrieve sulphur dioxide or the effective temperature of the

ozone layer with Brewers. Possible adaptations to spectral UV are also not excluded.

*Code and data availability.*    The original and the stray light-corrected Brewer direct-sun data, the `R` script for the radiative transfer simulations and the source code in `C++` for `rmstray` (software that corrects the count rates for stray light and saves the corrected values in a new B-file for use with any existing Brewer data analysis software) are available at https://doi.org/10.5281/zenodo.8097039 (Savastiouk and Diémoz, 2023).

**Appendix A:  Brewer working principles**

**A1    The Brewer spectrophotometer**

The Brewer system consists of a tracker, to turn the instrument towards the light source (generally the sun or the moon), and a spectrophotometer. The latter, in turn, is composed by the foreoptics, one or two monochromators, and a detector. The foreoptics include a zenith prism, whose rotation determines the elevation of the observing line of sight, and a set of filters. The

monochromator is a modified Ebert type that uses diffraction gratings to create a spectrum at the exit slits plane. The number of monochromators and the type of gratings determine the type of the Brewer:

- MkII Brewers are single-monochromator instruments with a 1800 lines $\mathrm{mm}^{-1}$ grating delivering the UV spectrum in the second order. MkV are just MkII Brewers with a modified filter wheel to be able to switch from UV to visible;

- MkIV Brewers are single-monochromator instruments with a 1200 lines $\mathrm{mm}^{-1}$ grating producing the UV in the third

order (and visible in the second order to retrieve nitrogen dioxide);

- MkIII Brewers have two monochromators and use 3600 lines $\mathrm{mm}^{-1}$ gratings, so that the UV spectrum is in the first diffraction order.

The gratings in all Brewer modifications can be turned with a high precision micrometer screw to select the portion of the spectrum that lands on 6 fixed exit slits. A single photomultiplier tube (PMT) is used as the detector.





To keep the wavelength setting in the Brewer stable, the diffraction grating is not turned during the direct-sun (DS) observations and the selection of the wavelengths is done instead by a fast-moving shutter that opens one exit slit at a time. Nominally, at the ozone operating position these exit slits are at 302, 306, 310, 313, 317 and 320 nm, with slight variability among the instruments. The convention is to number the exit slits 0 through 5. Slit 0 is mostly employed for wavelength calibration using an internal mercury lamp.

**A2   Standard Brewer direct-sun retrieval algorithm for ozone and sulphur dioxide**

The standard Brewer DS observation is a set of 5 individual measurements of solar radiation at the 6 slits plus a dark count reading. The standard Brewer algorithm first converts the detected counts into count rates by dividing the accumulated counts by the integration time. Next, count rates at each slit $j$ from 0 to 5, $I_{d_j}$ (the subscript $d$ stands for "detected"), are corrected for the dark count rate and linearity.

These count rates are converted into logarithmic space:

$$F_{d_j} = 10^4 \log_{10} I_{d_j} \tag{A1}$$

For reasons related to the limits in the computer memory at the time of the Brewer invention, the Brewer algorithm uses logarithm base 10 times $10^4$ and performs most operations after that with integer numbers. Using the Bouguer-Lambert-Beer's law (Bouguer, 1729) for radiation extinction in logarithmic form and with the assumption that only aerosol, air, ozone and

sulphur dioxide absorb/scatter strongly in the part of the spectrum that is measured by the Brewers we can write:

$$
\begin{aligned}
F_{d_j} = F_{0_j} \\
- (10^4 \log_{10} e)\,[\,\mu_{aerosol}\tau_{aerosol} + \mu_{air}\tau_{air} \\
+ \mu_{O_3}\tau_{O_3} + \mu_{SO_2}\tau_{SO_2}\,]
\end{aligned}
$$

(A2)

with $F_{0_j}$ being the logarithm base 10 of the count rates that would be detected outside of the Earth's atmosphere, $\mu$ the air mass factor needed to convert the slant column density (SCD) into the vertical column density (VCD), and $\tau$ being the optical depth for the corresponding compounds. In the Brewer algorithm, $F_{0_j}$ are explicitly corrected for the air (Rayleigh) scattering $\tau_{air}$ since it is easily calculated. $\mu$ is calculated with the current algorithm using the solar zenith angle at the time of

the measurement and the effective layer height of each compound.

    Linear combinations of $F_{d_j}$, i.e. $R_6$ and $R_5$, commonly called the second ratios since they are effectively logarithms of count rate ratios, are computed to make $R_6$ almost exclusively sensitive to ozone and $R_5$ almost exclusively sensitive to sulphur dioxide and ozone.





$$R_6 \quad = \quad -F_{d_2} + 0.5F_{d_3} + 2.2F_{d_4} - 1.7F_{d_5} \tag{A3}$$

$$R_5 \quad = \quad -F_{d_1} + 4.2F_{d_4} - 3.2F_{d_5} \tag{A4}$$

Note, that both $F_{d_1}$ and $F_{d_2}$ are used with the coefficient of -1. These are, as we mentioned earlier, the values mostly affected by the stray light. The effect of the stray light then is an underestimation of $R_6$ and $R_5$.

The ozone differential absorption coefficient in units $\text{cm}^{-1}$ for $R_6$, referred to as $A_1$ in the Brewer operating software, and the ratio of the sulphur dioxide absorption coefficient to the ozone absorption coefficient for $R_5$, referred to as $A_2$, are calculated as part of the instrument characterisation using the dispersion test results and laboratory-provided cross-sections for ozone and sulphur dioxide. Another differential absorption coefficient $A_3$ is calculated for $R_5$ using the ozone cross-sections to account for the sensitivity of $R_5$ to ozone. When computing the linear combination of the cross-sections, the result comes out negative, but out of convenience, $A_1$, $A_2$ and $A_3$ are defined as positive and, to compensate for that, the sign of the other terms in the Bouguer-Lambert-Beer's law is reversed as well. The extra-terrestrial coefficients for $R_6$ and $R_5$, i.e. $ETC_{O_3}$ and $ETC_{SO_2}$, are determined by a calibration process, as discussed later.

Now, the total ozone column $X_{O_3}$ in Dobson units (DU), can be calculated as

$$X_{O_3} = \frac{R_6 - ETC_{O_3}}{10\,A_1\,\mu} \tag{A5}$$

and the sulphur dioxide column $X_{SO_2}$ becomes

$$X_{SO_2} = \frac{R_5 - ETC_{SO_2} - 10\,X_{O_3}\,A_3\,\mu}{10\,A_2\,A_3\,\mu} \tag{A6}$$

The factor 10 accounts for the fact that the $F$'s were multiplied by $10^4$ and that 1 DU = $10^{-3}$ cm.

**A3 Calibration of the Brewer spectrophotometer**

The two commonly used methods for determining the ETCs for ozone and sulphur dioxide are the absolute calibration with the Langley plots (Langley, 1903) and a transfer of the calibration from another Brewer that has been already calibrated. Both methods are described in detail elsewhere (e.g., Redondas et al., 2018). Notably, in the calibration transfer method, two instruments take quasi-simultaneous DS measurements, then the ozone values from the reference (ref) instrument, $X_{O_3}^{ref}$, are used in Eq. (A5) for the instrument to be calibrated (new):

$$X_{O_3}^{ref} = \frac{R_6^{new} - ETC_{O_3}^{new}}{10\,A_1^{new}\,\mu} \tag{A7}$$

and $ETC_{O_3}^{new}$ is calculated. An average of these from a number of observations is used as the calibrated value. Assuming that the reference Brewer has no measurable stray light and ozone is calculated correctly from its measurements, the calibrated





ETC value will be underestimated if the Brewer to be calibrated has measurable stray light. This is important to remember when correcting for the stray light effect. Conversely, in an unlikely event where the reference Brewer has stray light and the new Brewer does not, then the ETC is overestimated.

**Appendix B: Details on simulations**

**B1    Input data**

In order to reproduce the Brewer behaviour, experimental data from the ordinary characterisation of a specific instrument are needed, i.e.

- Centre wavelength and resolution relative to each slit. These data, needed to simulate the core region of the instrumental bandpass function, are taken from the dispersion (DSP) test;

- Laser scans at 325 nm (Fig. 1), to simulate the amount and the effect of stray light in the shoulder and wing regions. 670 Details are provided in the next section;

- Brewer spectral sensitivity, to simulate the solar spectrum actually "seen" by the Brewer. The response through the direct port is here approximated by the UV response measured through the global entrance port (UV diffuser). There are small differences in the optical settings for the two observation geometries: the UV dome diffuser is not used for direct sun measurements and an internal ground quartz filter is added for direct sun, however both components are assumed to 675 be very neutral in colour. Everything else is the same in both observation geometries, including the components that introduce most of the spectral sensitivity observed in the Brewer, i.e. the photomultiplier detector, the UV combination filter, and to a lesser extent the neutral density filters.

    As highlighted by Kiedron et al. (2008), it should be noted that the UV response is normally measured with a lamp and that the response itself is affected by stray light. However, the same authors show that the effect is negligible in the 680 spectral range of interest.

**B2    Spectrum calculation**

We calculate the direct component of the solar irradiance reaching the Earth's surface, $I(\lambda)$, by the Bouguer-Lamber-Beer law accounting for ozone, sulphur dioxide and aerosols in the atmosphere. The configuration of the main parameters is reported in Table B1. The calculation of the spectrum extends over the wavelength range 280–400 nm. It is performed at high resolution 685 (0.005 nm) to accurately account for the spectral structures in both the solar spectrum and the ozone cross-section within the Brewer slit bands. We do not simulate solar irradiance at wavelengths longer than 400 nm as we assume that the Brewer responsivity is low in that range due to a combined effect from the PMT sensitivity dropping off and the UV combination filter, and that the contribution of out-of-range stray light is negligible (Pulli et al., 2018). Average atmospheric midlatitude conditions are employed for most of the simulations. In the base scenario, we set the sulphur dioxide VCD and AOD to zero,



**Table B1.** Main parameters employed in this work to simulate the solar spectra.

| Parameter | Value or bibliographic reference |
| --- | --- |
| Extra-terrestrial irradiance | Coddington et al. (2021) |
| Wavelength range (resolution) | 280–400 nm (0.005 nm) |
| Latitude | 45° |
| Pressure | 1000 hPa |
| Ozone VCD | 300 DU |
| Rayleigh scattering [a] | Bodhaine et al. (1999) |
| Ozone cross-section (297.5–332.4 nm) | Bass and Paur (1985), according to Redondas et al. (2014) |
| Ozone cross-section (outside the interval 297.5–332.4 nm) | Gorshelev et al. (2014) and Serdyuchenko et al. (2014) |
| Sulphur dioxide cross-section | McGee and Burris (1987) |
| Aerosol spectral extinction | Based on Ångström (1929) law |

[a] Only used to simulate the solar spectrum and the stray light spectrum, but not included when retrieving species, see details in Appendix B2

however these values were changed for specific tests. It should be noted that the model configuration, including the extra-terrestrial spectrum and the choice of the ozone and sulphur dioxide cross-sections, has a very limited impact on the simulated stray light effect, which mostly depends on the Brewer characterisation (laser line) and the ozone slant column.

We treat the effect of the stray light and the finite resolution of the instrument separately.

1. To simulate the stray light effect, we assume that measurements of the laser emission line at 325 nm outside the in-band
interval are representative of the Brewer stray light response to any input wavelength. Therefore, similarly to Zong et al. (2006), the scans are normalised to the integral within the in-band region $f_{IB}(\lambda)$ and this "core" region is set to zero, as we simulate the effect of the finite resolution in a different way. The resulting function, $f_{SL}(\lambda)$, is then convoluted with the solar spectrum to provide an estimate of the stray light spectrum $S(\lambda)$, i.e.:

$$S(\lambda) = \int I(\lambda')\, R(\lambda')\, f_{SL}(\lambda - \lambda')\, d\lambda' \tag{B1}$$

$R$ is the Brewer responsivity. The sign of the argument in $f_{SL}$ is due to the fact that the laser spectrum was obtained as a scan, i.e. the count rates recorded at wavelengths shorter than the laser peak are due to the stray light from longer wavelengths. The missing "right wing" of the scans in single Brewers (owing to the reduced spectral range of these instruments, only up to 328 nm) is unimportant, as solar radiation from shorter wavelengths does not significantly affect the count rates measured at longer wavelengths.

It should be highlighted that obtaining the stray light spectrum as a convolution in the wavelength space is only an approximation of the real stray light (some comments are given in Sect. S1 of the Supplementary materials). Thus, such



results should not be used to correct the Brewer data. Anyway, they provide a convenient framework to study the stray light behaviour with sufficient accuracy for our purposes.

Stray light can be switched off/on in the model depending on the considered case, i.e. S1 or S2–3 (Sect. 2.4). When we want to include the stray light effect, the term in Eq. (B1) is added to the direct irradiance spectrum weighted by the Brewer spectral response ($I(\lambda)\,R(\lambda)$), otherwise only the latter product is considered;

2. To simulate the finite resolution of the Brewer at the various slits, the high-resolution spectrum obtained from the previous step is further integrated over each slit function. These latter are approximated as trapezoids, i.e. isosceles triangles cut at 87 % of their height, according to Wardle (2001) (also Moeini et al., 2019), as normally done when processing the results of the dispersion test;

3. To correct for stray light in the synthetic spectra (S3), a fraction of the simulated count rates at the longest wavelength (slit 5) is subtracted from the count rates of the other slits (notice that this operation must be performed before calculating the logarithm base 10).

## B3   Synthetic retrievals

After simulating the solar irradiance measured by the Brewer, the logarithms of the (corrected or uncorrected) count rates are linearly combined as in the traditional Brewer algorithm (Appendix A2), using the standard weighting coefficients.

The effect of molecular scattering along the atmospheric optical path is here compensated by correcting the linear combination using the same Rayleigh cross-sections employed to calculate $I(\lambda)$ in place of using the standard Brewer Rayleigh correction. This avoids systematic errors due to the incomplete removal of the Rayleigh scattering in Brewers (Carlund et al., 2017), to focus on the stray light effect only. The ETCs of the virtual Brewer must be determined in order to retrieve $O_3$ and $SO_2$. In simulation S1 (Sect. 2.4), they are obtained by iterating the simulations at several air masses and using the Langley extrapolation method, similarly to what is usually done with real-world reference instruments, assuming they have no stray light. In simulations S2–S3, the ETCs are obtained by transfer based on the a-priori VCDs given to the model. The absorption coefficients $A_1$, $A_2$ (this one set to a fixed value of 2.35, in the standard Brewer algorithm) and $A_3$ are recalculated for each simulated instrument based on the cross-sections listed in Table B1 and the experimental characterisation (measurement wavelengths and resolutions).

*Author contributions.*  VS and HD conceived the presented idea and designed the methodology. VS processed the experimental data and HD performed the radiative transfer calculations. VS and HD prepared the initial draft of the manuscript. CTM provided insight to better understanding of the instrument and its physics, carefully reviewed and contributed to the final manuscript.

*Competing interests.*  The authors declare that they have no conflict of interest.



*Acknowledgements.* The authors acknowledge the National Oceanic and Atmospheric Administration (NOAA) and Environment and Climate Change Canada (ECCC) for the data from their Brewers (Table 1). The authors are very grateful to T. Grajnar and M. Brohart who performed the laser scans in Fig. 1. H. Diémoz was partly supported by the European Space Agency – European Space Research Institute (ESA/ESRIN) in the frame of the Quality Assurance for Earth Observations (QA4EO) contract.



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
