# Peer review of "A physically-based correction for stray light in Brewer spectrophotometer data analysis"

_EGUsphere, 2023_

## Referee Comment (RC1)

Review of the manuscript "A physically-based correction for stray light in Brewer spectrophotometer data analysis" by V. Savastiouk et al.

The paper describes an easy to use new procedure to take into account spectral stray light in the derivation of total column ozone and total sulphur dioxide from measurements of solar irradiance by single Brewer spectrophotometers.

The method uses a single parameter to quantify the impact of stray light on total column ozone (and a second parameter for the SO2 retrieval), which is specific to a Brewer spectrophotometer. These parameters are usually retrieved by comparison to a reference instrument, that is not sensitive to stray light (e.g. a double Brewer for example, or an already corrected single Brewer), even though a method is explained how these parameters can be retrieved without external reference instrument. A detailed uncertainty budget is also provided.

The method is validated by applying it to several single Brewer instruments, that are collocated with a double Brewer used as the reference. Furthermore, a theoretical justification of the stray light correction methodology is provided by modelling the stray-light impact from laboratory based slit function measurements for one Brewer instrument (Brewer #009).

The authors are well-known scientists, and experts of the Brewer spectrophotometers. The paper is very well written and understandable to a broad audience (in my opinion). Prior work is mentioned as needed, and the references are comprehensive and complete. Figures, tables are all of excellent quality.

I commend the authors for this paper that will definitely have an impact on the data quality of the Brewer network.

In my opinion, the paper can be accepted as is, even though I list a number of points below which the authors might want to take into account, or at least answer in their response to this review:

1) The authors claim a "physically-based correction for stray light". I assume that they base this claim on the fact that they introduce a stray-light correction concept based on slit functions, and validated through a physical model of the Brewer. However the validation is really only qualitative, since the retrieved stray-light factor alpha is off by a factor of 2 when comparing the value obtained from the simulation with the actual factor obtained from the outdoor comparison using real measurements. The authors claim that the discrepancy comes from using outdated slit function measurements of this particular Brewer (#009), and that in the meantime this Brewer has changed. The comparison between simulation and measurements would have been more convincing if the authors would have been able to use a Brewer where the slit functions are known and could be trusted. In the current version of the manuscript, the discrepancy leaves the argument hanging. Without a better validation, this method is also only sort of empirical, where the physical justification is well argued, but not substantiated.

2) line 304: "orthogonality between alpha and ETC": Without clear definition of this term, it sounds a bit colloquial, and I suggest to replace it by a more precise statement. For example the term "correlated" or "uncorrelated" could be used here.

3) Line 318 : "the results show that Langley plots with measurements affected by stray light make little sense" is a worrying statement, considering that historically, the Brewer network traceability was based on a triad of single Brewers. Could the authors comment on the potential impact to the historical datasets of the Brewer network, and if it would be necessary and possible to apply this stray light correction procedure retrospectively to the whole network?

4) table 3 . it would be helpful to briefly describe the terms of the first column in the table caption. For example I do not understand what DeltaETCO3 stands for.

5) In the uncertainty budget, Table 5, the contribution labelled "Mathematical model", is probably more correctly named "residuals", as the residuals between retrieving the model parameters by fitting the model to the measurements?

6) the discussion S1 in the Supplement, and referred to at lines 705-708 on the correctness of using measured slit functions as a basis for a stray-light correction is not very convincing. The authors are correct in stating that the stray light measured through the various slits of the Brewer spectrophotometer can differ between themselves, and that the common orientation of the grating could produce a common feature in the slit functions. However this does not mean that there is some additional mysterious effect that could produce additional stray light that would not be captured by an accurate slit function measurement. To avoid any misunderstanding, I would recommend to rewrite this part of the manuscript (or to skip as it has no impact on the paper itself).

---

## Author Comment (AC1)

Review of the manuscript "A physically-based correction for stray light in Brewer spectrophotometer data analysis" by V. Savastiouk et al.

The paper describes an easy to use new procedure to take into account spectral stray light in the derivation of total column ozone and total sulphur dioxide from measurements of solar irradiance by single Brewer spectrophotometers.

The method uses a single parameter to quantify the impact of stray light on total column ozone (and a second parameter for the SO2 retrieval), which is specific to a Brewer spectrophotometer. These parameters are usually retrieved by comparison to a reference instrument, that is not sensitive to stray light (e.g. a double Brewer for example, or an already corrected single Brewer), even though a method is explained how these parameters can be retrieved without external reference instrument. A detailed uncertainty budget is also provided.

The method is validated by applying it to several single Brewer instruments, that are collocated with a double Brewer used as the reference. Furthermore, a theoretical justification of the stray light correction methodology is provided by modelling the stray-light impact from laboratory based slit function measurements for one Brewer instrument (Brewer #009).

The authors are well-known scientists, and experts of the Brewer spectrophotometers. The paper is very well written and understandable to a broad audience (in my opinion). Prior work is mentioned as needed, and the references are comprehensive and complete. Figures, tables are all of excellent quality.

I commend the authors for this paper that will definitely have an impact on the data quality of the Brewer network.

In my opinion, the paper can be accepted as is, even though I list a number of points below which the authors might want to take into account, or at least answer in their response to this review:

We thank the reviewer for taking the time to revise our manuscript and for their constructive comments.

Our point-to-point reply is given hereafter.

1) The authors claim a "physically-based correction for stray light". I assume that they base this claim on the fact that they introduce a stray-light correction concept based on slit functions, and validated through a physical model of the Brewer. However the validation is really only qualitative, since the retrieved stray-light factor alpha is off by a factor of 2 when comparing the value obtained from the simulation with the actual factor obtained from the outdoor comparison using real measurements. The authors claim that the discrepancy comes from using outdated slit function measurements of this particular Brewer (#009), and in the meantime this Brewer has changed. The comparison between simulation and measurements would have been more convincing if the authors would have been able to use a Brewer where the slit functions are known and could be trusted. In the current version of the manuscript, the discrepancy leaves the argument hanging. Without a better validation, this method is also only sort of empirical, where the physical justification is well argued, but not substantiated.

The reviewer's point is noted, however we stand by our assertion that PHYCS is physically-based. We call our algorithm physically-based because the mathematics of it is based on the physical principle behind the instrumental internal stray light, namely the additional, "stray", light that contributes to the detected count rate at every measurement. Also, in contrast with other attempts to correct for the stray light in the Brewers, our algorithm only uses measured quantities, i.e. the count rates detected by the counting system. The algorithm does not introduce "external" or calculated values like the air mass factor as many other proposed corrections do. As such, we believe calling our algorithm physically-based is justified.

The absolute values for the correction factors were not the main result of our modelling of the stray light in the Brewers. The most important takeaway from the simulations is the finding that the stray light contribution, in count rate space, is relatively constant across the measured wavelengths, making it possible to apply it in a simple way. This is because the largest contribution to the stray light is coming from longer, less affected by ozone and thus much brighter wavelengths. By applying this correction to experimental data we confirmed that this is working very well.

The laser scan that we used for Brewer #009 shows the "wings" at a level of $10^{-4}$. An offset of $10^{-4}$ was added to the laser scan. With that small addition in absolute sense, the model predicts parameter alpha to be 0.4%, same as our experimental data shows. Please also refer to the response for your point 6), where we describe other possible shortcomings in the laser scanning process.

2) line 304: "orthogonality between alpha and ETC": Without clear definition of this term, it sounds a bit colloquial, and I suggest to replace it by a more precise statement. For example the term "correlated" or "uncorrelated" could be used here.

We agree with the reviewer. Modified: "Since alpha and ETC are uncorrelated, the two unknowns can be determined independently without significant cross-talks"

3) Line 318 : "the results show that Langley plots with measurements affected by stray light make little sense" is a worrying statement, considering that historically, the Brewer network traceability was based on a triad of single Brewers. Could the authors comment on the potential impact to the historical datasets of the Brewer network, and if it would be necessary and possible to apply this stray light correction procedure retrospectively to the whole network?

Thank you for this important comment. The accepted uncertainty of Langley method when applied to Brewers, is +/- 5 units in ETC (roughly +/- 1.5 DU when solar zenith angle is 0 and less when sza>0). We estimate that the error in ETC from the Langley method due to stray light is 10 to 15 units, making this contribution significant enough to be of concern when discussing the quality of calibrations of the primary standards, but still its effect on the ozone data is less than 1% at air mass factors > 1.5, which most of the Brewer data are. We do not see this error warranting recalculation of past absolute calibrations of the primary standards. However, as we mention in the paper, calibration transfers to field instruments were significantly affected by stray light and definitely deserve attention of the Brewer community.

What we wanted to stress by our statement, is that now that a simple algorithm for correcting for stray light exists, there is no justification for not applying it to the data, including those data that are used for the Langley method calibrations. We have replaced the wording "makes little sense" with "lead to errors in the determination of the ETC".

It is indeed not only possible, but also imperative to reprocess past calibration data and then apply our algorithm to past data from all Brewers. While our algorithm is easy to implement, the sheer volume of the Brewer data makes this task both important and time-consuming, requiring additional resources to do it right. We would like to also stress that not all data centres are able to track/identify changes in the submitted data, which will lead to challenges for the data end users when some data have been corrected and some have not.

A comment about the need for calibrations and data reprocessing has been added to the Discussion section.

4) table 3 . it would be helpful to briefly describe the terms of the first column in the table caption. For example I do not understand what DeltaETCO3 stands for.

We appreciate your pointing this out. The following text was added in the table caption: "The rows report, respectively: the serial number of the reference MkIII Brewer, the $\alpha$ and $\beta$ coefficients used in the correction, the ozone extra-terrestrial coefficient after the application of the stray light correction, the original extra-terrestrial coefficient used in the Brewer, the difference ($\Delta$ETC) between the two values above, and the minimum air mass factor available in the analysis."

5) In the uncertainty budget, Table 5, the contribution labelled "Mathematical model", is probably more correctly named "residuals", as the residuals between retrieving the model parameters by fitting the model to the measurements?

"Mathematical model" is the wording used by the Guide to the expression of uncertainty in measurement (GUM; BIPM et al., 2010). The word "residuals" has been added in brackets to avoid misinterpretation.

6) the discussion S1 in the Supplement, and referred to at lines 705-708 on the correctness of using measured slit functions as a basis for a stray-light correction is not very convincing. The authors are correct in stating that the stray light measured through the various slits of the Brewer spectrophotometer can differ between themselves, and that the common orientation of the grating could produce a common feature in the slit functions. However this does not mean that there is some additional mysterious effect that could produce additional stray light that would not be captured by an accurate slit function measurement. To avoid any misunderstanding, I would recommend to rewrite this part of the manuscript (or to skip as it has no impact on the paper itself).

We appreciate the reviewer's position and would like to clarify ours. The issue here is that in order to have "accurate slit function" it needs to be accurately and precisely measured for each of the 6 slits used in the Brewer and it has to be measured at the operating wavelength at each of the slits, which can only be done using a tunable laser as we state in the paper. The scan should preferably be made with a single pass for a slit. However, the dynamic range of the Brewer counting system cannot cover a factor of $10^9$ that is needed for an accurate scan of a laser line peak and the wings at the same attenuation level, even if a tunable laser is available.

Our comment about some features that correspond to a particular diffraction grating angle is aimed to stress the sometimes significant differences between instruments, when some Brewers potentially have reflective surfaces that contribute to stray light, but are difficult to characterise in a model. While this effect is not mysterious, and has been detected in several instruments, it poses challenges that are not easily overcome in modelling. PHYCS, when applied to experimental data, works well for Brewers that show this effect, which is what is important in practice.

---

## Author Comment (AC2)

This manuscript presents a highly innovative method of correcting stray light derived errors in total column amounts of ozone and sulfur dioxide measured by the single Brewer spectrophotometers. Different from the certain correction methods, the presented has very clear  basis of physics consideration and more feasible application for the Brewer ozone spectrophotometer community. Furthermore, this physics-based method is also validated by theoretical simulations. The manuscript has a very significant application value for the single Brewer spectrophotometer. All the authors of the manuscript are well-known scientists who have long been engaged in works of Brewer spectrophotometer. This manuscript is recommended to be published with small revisions. There are following two comments:

We thank the reviewer for taking the time to revise our manuscript and for their constructive comments.

Our point-to-point reply is given hereafter.

1) Line # 278, "The ozone ETC, in simulation S3, strictly agree with the one observed without stray light in the model (2583) "---What is the meaning of 'model (2853)'?

Thank you for pointing this out. The text has been modified: "The value of the ozone ETC in simulation S3, i.e. 2583, matches the one obtained without stray light in the model".

2) Given the significant role of travelling standard # 017 in history, a recommendation of reprocessing the historical data measured by those single Brewer spectrophotometers which had been periodically calibrated by #017 had better been added in "3.2.5 Changes in stray light with time"   when Table IV is shown.

We appreciate this important comment. Added the following text to the discussion section: "5. Past calibrations when either the reference or the field Brewer, or both, were instruments affected by stray light, need to be reprocessed using PHYCS and then the data from the field Brewers reanalyzed. "

---

## Author Comment (AC3)

**Review of Savastiouk et al. , A physically-based correction for stray light in Brewer spectrophotometer data analysis**

**General comments**

The submitted manuscript describes a new method for correcting stray light effects for single monochromator Brewer spectrophotometers.

The authors propose a very simple parameterization which they justify using calculations from a raditive transfer model and multiple examples of observational data, with quite impressive results. The new algorithm has been implemented in software available from the authors.

The topic is a very important one for the atmospheric measurement community, the manuscript is clearly written and in my opinion is suitable for publication in AMT with only minor revisions.

An appendix has been included describing the working principles of the Brewer as needed to follow the rest of the paper, which is very helpful.

At times the language does verge into something more like an advertisement (eg lines 530-538) rather than a sober and objective scientific paper.

Lots of interesting discussion has been included and the authors clearly have spent a lot of time thinking deeply about Brewers, but at times some of this discussion is probably to the detriment of the overall focus and impact.

The arguments presented seem quite sound to me and convincing, but before the method is widely adopted in the global Brewer network, I would hope to see some careful comparisons of the results obtained compared to those from other methods, in particular the Redondas et al. technique which I believe  is now standard in EUBREWNET.

A final general comment is that the authors seem to intermingle the discussion of the algorithm itself with their own software implementation of it, which I found confusing at some points. In principle, the algorithm and the software are distinct. They seem to refer to both as "PHYCS", although this name does not appear in the abstract.

My comments are mostly very minor.

We thank the reviewer for taking the time to revise our manuscript and for their constructive comments.

The acronym PHYCS has been added to the abstract.  There are only very limited places in the text where there is a reference to the software, and those now use the name of the software, "rmstray".

Commenting on a possible careful comparison with other stray light correction methods is outside the scope of this work. The paper lists what we see as PHYCS advantages. In the publication that introduced the correction currently used in EUBREWNET (Redondas et al., 2018) significant shortcomings of the method are mentioned for data at slant ozone larger than 1800 DU where the corrected data were not within 1% of the reference MKIII. In contrast, PHYCS results in data well within 1% for all analysed data up to 2200 DU of ozone in the path. In a private communication with EUBREWNET personnel, we have been informed that they are implementing PHYCS to replace any currently used stray light corrections.

Our point-to-point reply is given hereafter.

**Specific comments**

Line 5 You say "virtually eliminating" but then only shortly later (line 21) you say you can see it. In practice your method relies on assuming the comparison MKIII has no stray light it seems to me?

Thank you for this comment. The double Brewer has an extremely low contribution of stray light, the lowest among the ground-based ozone spectrophotometers. This makes the use of the MKIII as the reference justified. There is some evidence - but not conclusive - that some small effect may still be here in the ozone measurements. And correcting single Brewers to agree with the Doubles is a big step forward. Lines 20-21 were removed from the abstract to avoid any confusion.

Line 8 Add a word such as "contribution" or "effect" after "a small"

Agreed. Added.

Line 9 Please reword – stray light doesn't reduce the "slant ozone" itself of course.

Good point. Modified: "This is because even a small additional stray light contribution at shorter wavelengths significantly reduces the calculated slant column density at large values."

Line 15 The reader can't know what "matching ozone calculations from the single and double mono-chromator Brewers" means, please re-word for better clarity, ie side-by-side comparison

We appreciate this comment. Modified to say: "… Brewers making measurements side-by-side."

Lines 18-21 I suggest removing these lines from the abstract. Lines 18-22 describe further work, and 20-21 is a very tentative result.

The position of the reviewer is noted and partially implemented.

As stated in the paper, correcting for the stray light in the count rate space, the data can be used for any further processing with ozone and sulfur dioxide being simply the most common products from the Brewer data.

The sentence about tentative results assessing stray light effects in the MKIII Brewers was removed from the text.

Line 29 Why do you specifically mention NDACC here? I count only eight Brewer stations on the NDACC website! Wouldn't WMO-GAW or EUBREWNET be more appropriate?

This is indeed an important point. The reference to NDACC was replaced by "an essential component of the WMO-GAW O3 observing system "

Line 52 Only true for the northern hemisphere, and not Antarctica.

Respectfully disagree. As shown in the paper, the stray light affects the ozone data at as little as 600 DU SCD. With the minimum solar zenith angle being larger in the winter in mid-latitudes in both hemispheres, even relatively small TOC will lead to SCD over 600 DU.

Line 57 Remove "the" in "the space"

Noted. Removed.

Line 67 Is "PHYCS" the algorithm or the software?

There are only very limited places in the text where there is a reference to the software, and those now use the name of the software, "rmstray".

Line 67 How do the authors want "PHYCS" to be pronounced?! ("Ficks"?)

Although it seems rather obvious , 'pronounced as "*fix*" ' was added to the text for clarity.

Line 71 I don't think you mean "peculiar" here.

Indeed. Changed.

Lines 78-79 Wouldn't this advantage also be the case for other available stray-light correction methods?

Possibly. But the statement is still appropriate regarding PHYCS.

Table 2  For consistency, Hobart should be in Australia – for the other sites you have given the name of the country.

Agreed.  Corrected.

Table 2 I suggest putting the stations in the table in order of latitude or magnitude of latitude, so that the reader can more easily grasp the "vastly different observing conditions" which you claim.

Thank you for this helpful suggestion.  Implemented ordering by latitude.

Lines 106-113 Personally I found the construction of this run of sentences quite repetitive, all beginning the same way,  "Additionally,", "Indeed,", "Additionally" and "In fact,".

We appreciate your taking the time to read the manuscript this closely.  The language has been smoothed over.

Figure 1  The figure is quite important for the paper but it seems strangely orphaned, without much description of where it has come from. Is there a reference?  (It sounds like it was work done twenty years ago but never written up?)

Thank you for this important comment. This work has been presented at several meetings.  A reference to the Brewer User Workshop in Seoul in 2007 has been added.

Line 142 Shouldn't what you have called here $I_{320}$ be written as $I_d(320)$ to follow the notation of equation (1)?

Yes.  Corrected.

Line 144 At this point, alpha is not a constant, but a function of the wavelength and presumably other parameters to describe the atmosphere.  The hypothesis coming next is that alpha in fact doesn't depend on the wavelength or the conditions.

This has been recognized as a writing issue and corrected.

Lines 146-149 I find this confusing because here you seem to be saying there is one value of alpha for slits one to five but shortly afterwards you introduce beta for slit 1 and later alpha only actually seems to ever be used for slit 2 anyway?

This is described in the manuscript.  There is a wavelength separation from the 4 slits used for ozone measurement and the one which the SO2 measurement is most dependent on. That is the reason for two different coefficients.  The text has been modified with a better formulation.

Line 170 Add "in practice" or similar wording after "In most cases"

Agreed.  Done.

Lines 170-172 I would say an absolute calibration can be performed wherever you like but the results might not be very good, so add "to the desired uncertainty" or words like that here.

The reviewer is correct. Added "to the desired uncertainty"

Line 200 Change to either "in the tropics" or "close to the equator"

Noted and modified: "are close to the equator"

Lines 2011-2014 But wouldn't other problems also present the same symptoms, for example any change in the calibration?

We appreciate the reviewer's attempt to propose other possible problems that can be considered. The practical problems have been stated in the text, others were left out.

Line 220 "first" not "fist"

Indeed. Corrected.

Figure 2 You should label the dashed lines in both (a) and (b) clearly with the slit numbers, this would make the text much easier to follow.

Thank you for this helpful comment. Figure 2 now has slit numbers marked.

Line 276-278 I would tone the language down a little bit here, ie "impressive", "especially", "perfectly" .

Noted and adjusted.

Lines 279-282 Yes this is a good point to mention, I was wondering about whether high aerosol optical depth (in the UV) would have an effect, possibly depending on its spectral properties. Is this worth discussing?

This discussion would be outside the scope of the paper. The text clearly states that stray light sources other than internal instrumental spectral effects would not be addressed.

Line 288 I don't think you mean "likely" here?

The word "likely" has been removed from the text.

Lines 336-338 Just to be clear here, you are using the Mk III to derive alpha and then comparing the new results with the same Mk III instrument again?

The short answer is Yes. A more detailed answer is provided in the paper where it is stated that the derivation of the correction factors is done based on one day of measurements, then the comparison is done over 6 months making the datasets for "calibration" and "verification" independent.

Line 351 Are you worried that these "simplified assumptions" and "slight discrepancies" have such a big effect?

Judging the size of the assumptions' effects on modelling is a tricky business. Even though the model is not perfect, it provides understanding of the physical processes and their implications on the data. The important part is that PHYCS works on the experimental data.

Addressing another reviewer's question, the following text has been added: "The laser scan that was used for Brewer #009 shows the "wings" at a level of $10^{-4}$. As a test, an offset of $10^{-4}$ was added to the laser scan. With that small addition in absolute sense, the model predicts parameter alpha to be 0.4%, same as in the experimental data."

Line 353 Insert "the" before "year"

Noted and added.

Lines 424-470 I'm pleased that you have included the discussion of uncertainties. Everything you say seems quite reasonable to me, but I wouldn't be surprised if later work modified some of the conclusions to some extent.

Pleased to see that our arguments resonated with you.

Lines 471 – 493 I am not sure of the value of section 3.2.7. For this method to be applied in the network you need to assume the Mk IIIs have no stray light, don't you?

Respectfully, No. It is a major improvement to upgrade the calibration of the singles to agree with doubles. There is always more to do and it is not clear whether the small discrepancies in the MKIIIs are actually stray light.

Lines 486-493 This section seems to be almost thinking aloud, and I suggest deleting it. Even if you could perform this experiment, what specific indications would you be looking for as a signature of stray light, rather than some other issue?

These lines contained very important points. However, after a careful consideration, this section has been removed.

Line 506 Delete "to"

Done

Lines 502-512 This is quite an interesting discussion, but it would be better with some numbers. The situation of a Brewer being calibrated at low latitude but then being operated at high latitudes must have happened a lot over the last thirty years.

Thank you for pointing this out. This has indeed happened many times. The following paragraph, with numbers, has been added to provide an example for this effect.

"For example, the data from Brewer #009 at MLO show ozone underestimation approaching 1 % at slant ozone of approximately 800 DU (Fig. 5a). If this instrument were to be moved to Alert, where the minimum measured slant ozone amount is just less than 800 DU (Fig. 5e), then with the current calibration the ozone VCD would be underestimated all year round. In contrast, Brewer #029, having its calibration done on site in Alert, provides the data to within 1 % up to 1200 DU of ozone in the path."

Line 513 "A few word but an important addition" .. please re-word

Thank you for noticing this. Done

Lines 530-538 This reads like a salesman's pitch! Please re-word to be more specific and informative.

Agreed. The whole section has been rewritten to be more concise and more informative.

It now reads: "In summary, once PHYCS has been implemented, the Brewer network will be able to consistently provide reliable data at large slant ozone columns. At high latitudes, this will translate to more data availability throughout the year. A high quality of Brewer calibrations performed on-site, especially at mid and high latitudes, will be easier using a more portable single-monochromator Brewer. Consistency of the data quality collected with the Brewers of different ages (and thus different stray light contributions) will be significantly improved."

Line 569 "Effortlessly" is excessive

Replaced with "easily"

Line 640 I would like to see this point expanded just slightly, to explain the combination of the laboratory cross-section and the particular slit function

Added: "Absorption coefficients are calculated as a linear combination of cross-sections that have been convolved with the measured slit functions. Currently, cross-sections are used at a constant effective ozone temperature of -44∘C."